# Thalamic reticular nucleus induces fast and local modulation of arousal state

**Laura D Lewis[1,2,3†], Jakob Voigts[3†], Francisco J Flores[3,4,5], L Ian Schmitt[6], Matthew A Wilson[3,7], Michael M Halassa[6*‡], Emery N Brown[3,4,5,8,9*‡]**

[1]Society of Fellows, Harvard University, Cambridge, United States; [2]Athinoula A. Martinos Center for Biomedical Imaging, Massachusetts General Hospital, Boston, United States; [3]Department of Brain and Cognitive Sciences, Massachusetts Institute of Technology, Cambridge, United States; [4]Department of Anesthesia, Harvard Medical School, Boston, United States; [5]Department of Anesthesia, Massachusetts General Hospital, Boston, United States; [6]Neuroscience Institute, New York University, New York, United States; [7]Picower Institute for Learning and Memory, Massachusetts Institute of Technology, Cambridge, United States; [8]Harvard-MIT Health Sciences and Technology, Massachusetts Institute of Technology, Cambridge, United States; [9]Institute for Medical Engineering and Science, Massachusetts Institute of Technology, Cambridge, United States

**\*For correspondence:** michael.halassa@nyumc.org (MMH); enb@neurostat.mit.edu (ENB)

[†]These authors contributed equally to this work
[‡]These authors contributed equally to this work

**Abstract** During low arousal states such as drowsiness and sleep, cortical neurons exhibit rhythmic slow wave activity associated with periods of neuronal silence. Slow waves are locally regulated, and local slow wave dynamics are important for memory, cognition, and behaviour. While several brainstem structures for controlling global sleep states have now been well characterized, a mechanism underlying fast and local modulation of cortical slow waves has not been identified. Here, using optogenetics and whole cortex electrophysiology, we show that local tonic activation of thalamic reticular nucleus (TRN) rapidly induces slow wave activity in a spatially restricted region of cortex. These slow waves resemble those seen in sleep, as cortical units undergo periods of silence phase-locked to the slow wave. Furthermore, animals exhibit behavioural changes consistent with a decrease in arousal state during TRN stimulation. We conclude that TRN can induce rapid modulation of local cortical state.

## Introduction

Modulation of arousal is one of the central aspects of behavior, as sleep plays an essential role in cognitive function and survival. A key marker of decreased arousal is cortical slow wave activity (1–4 Hz), which occurs both during non-REM sleep (*Vyazovskiy et al., 2009*; *Amzica and Steriade, 1998*; *Buzsaki et al., 1988*) and in awake animals during low vigilance states and sleep deprivation (*Vyazovskiy et al., 2011*; *Huber et al., 2000*). The slow wave marks rhythmic periods of suppression in cortical neurons (OFF periods) lasting hundreds of milliseconds (*Vyazovskiy et al., 2009*; *Steriade et al., 2001*). These brief offline periods are a candidate mechanism for decreased arousal, and slow waves in local cortical regions are associated with behavioral deficits on sub-second time-scales (*Vyazovskiy et al., 2009*). Slow waves are thus correlated with both behavioral decreases in arousal and disruption of cortical activity. However, while several brainstem structures for global control of sleep states have been well characterized (*Giber et al., 2015*; *Tsunematsu et al., 2011*; *Adamantidis et al., 2007*; *Anaclet et al., 2014*), no mechanism has been identified that generates the spatially isolated slow waves that occur during drowsiness, known as 'local sleep'. Slow wave activity is locally regulated both during sleep, where it plays a role in sleep-dependent memory

**eLife digest** We usually think of sleep as a global state: that the entire brain is either asleep or awake. However, recent evidence has suggested that smaller regions of the brain can show sleep-like activity while the rest of the brain remains awake. It is not clear why or how these sleep-like patterns of brain activity appear, and whether they are related to the drowsy behaviour that occurs when one is about to fall asleep.

Lewis, Voigts et al. investigated how this process works in mice using a technique called optogenetics. This technique makes it possible to genetically engineer mice so that the activity of particular areas of the brain can be switched on or off by light. Lewis, Voigts et al. used light to stimulate different regions of the brain and tracked the resulting brain activity using tiny electrodes that are capable of detecting the activity of individual neurons.

The experiments show that stimulating one part of a deep brain structure called the thalamic reticular nucleus causes just one small part of the brain to switch from being awake to producing sleep-like brain wave patterns. When a larger area is stimulated, the whole brain switches into this sleep-like pattern. Stimulation of the thalamic reticular nucleus also caused the animals to become drowsy and they were more likely to fall asleep, which suggests that sleep-like activity in small parts of the brain may contribute to drowsiness.

Lewis, Voigts et al.'s findings identify a brain switch that can influence whether an animal is awake or asleep. Importantly, they show that sleep can be independently controlled in small brain regions, and that the thalamic reticular nucleus contains a 'map' that allows it to induce sleep in just one region, or across the whole brain. Memories are strengthened during sleep, so the next challenge is to study whether the thalamic reticular nucleus influences memory formation. The findings also suggest that further study of this brain region may be useful for understanding how the sleep and awake states are controlled by particular neurons.

consolidation (*Huber et al., 2004*), and in the awake state, where it reflects a shift in cortical processing modes (*Wang et al., 2010*). Local modulation of slow waves is therefore an important element of cortical function, but the underlying mechanism is not well understood.

We sought to identify a forebrain structure that modulates local cortical slow wave activity. A central modulator of corticothalamic feedback that could initiate these dynamics is the thalamic reticular nucleus (TRN), a subcortical structure that provides powerful inhibition to dorsal thalamic nuclei. The TRN is a thin sheath of GABAergic neurons that surrounds the thalamus and inhibits thalamic relay cells (*Pinault, 2004*; *Guillery and Harting, 2003*). TRN has been implicated in sensory processing (*Hartings et al., 2003*; *Deleuze and Huguenard, 2006*), attentional gating (*McAlonan et al., 2008*; *Crick, 1984*; *Halassa et al., 2014*; *Wimmer et al., 2015*), and sleep state modulation (*McCormick and Bal, 1997*; *Steriade, 2000*) - and is uniquely positioned to selectively and rapidly modulate cortical state. TRN has a causal role in initiating sleep spindles (*Halassa et al., 2011*; *Barthó et al., 2014*; *Bazhenov et al., 2000*; *Steriade et al., 1987*), and molecular genetic manipulation of TRN conductances reduces EEG sleep rhythms (*Cueni et al., 2008*; *Espinosa et al., 2008*), indicating a role for thalamocortical feedback in cortical sleep oscillations. However, direct manipulations of thalamic activity have yielded conflicting results. Nonspecific activation of multiple thalamic nuclei (including TRN) increases time spent in sleep (*Kim et al., 2012*), whereas selectively stimulating thalamus induces a desynchronized cortical state (*Poulet et al., 2012*), suggesting a role for thalamus in controlling arousal states. On the other hand, directly disrupting thalamic activity does not induce slow waves or sleep states (*Constantinople and Bruno, 2011*; *Steriade et al., 1993*; *David et al., 2013*). These mixed findings suggest a complex involvement of thalamus in regulating behavioral arousal, which could be mediated through the TRN. In addition, many sedative and anesthetic drugs act to enhance GABAergic synaptic transmission and thus potentiate the effects of TRN activity (*Brown et al., 2011*), further suggesting that it could be a component of the mechanism by which these drugs induce an unconscious state (*Franks, 2008*). Neuronal activity in TRN is known to correlate with arousal (*Halassa et al., 2014*; *Barrionuevo et al., 1981*; *Steriade et al., 1986*), but it remains unclear whether these firing patterns are a consequence of low arousal or a cause. In particular, the role of TRN in generating the low-frequency oscillatory dynamics characteristic of low

arousal states is not known, and the behavioral significance of such cortical dynamics has not been causally tested.

Here, we optogenetically activated TRN, and found that this manipulation rapidly induces local sleep-like thalamocortical slow waves. Tonic activation of TRN in awake animals produced slow wave activity in the associated cortical region, together with phase-locked periods of silence in cortical neurons (OFF periods). This manipulation also produced a progressive decrease in arousal state: awake animals exhibited less motor activity and spent more time in non-REM sleep, and anesthetized animals exhibited a decrease in cortical activity and a shift in dynamics favoring OFF periods. We find that the net effect of TRN stimulation is to decrease thalamic firing, suggesting that TRN may modulate arousal state through selective inhibition of thalamic activity, facilitating the onset of slow waves. Furthermore, TRN and other thalamic neurons are phase-locked to the induced oscillations, suggesting that TRN, thalamus, and cortex are all engaged in the rhythm. We conclude that tonic depolarization of TRN rapidly modulates cortical state and controls the animals' arousal, by inducing suppression and rhythmic spiking in thalamus. The spatial characteristics and rapid timescale (<50 ms) of these effects show that local oscillatory dynamics between thalamus and cortex are a central mechanism for modulation of arousal.

## Results

We sought to identify a structure that modulates local cortical slow wave activity in a rapid and spatially restricted manner. To optogenetically manipulate the TRN, we first used transgenic mice in which channelrhodopsin2 (ChR2) expression was under the control of the vesicular GABA transporter (VGAT-ChR2) (*Zhao et al., 2011*). In these mice, the TRN exhibits preferential expression of ChR2 compared to surrounding subcortical regions (*Figure 1—figure supplement 1–2*) (*Halassa et al., 2011*), allowing us to manipulate TRN activity in awake mice using chronically implanted optical fibers (*Figure 1a*).

### Tonic activation of TRN produces cortical slow waves

We first tested how tonic activation of TRN affected neural dynamics in the cortex of awake head-fixed mice. We implanted four mice with stereotrodes distributed across cortex and an optical fiber targeting the somatosensory sector of TRN (*Figure 1a*). To examine the intrinsic cortical dynamics that emerge when no specific oscillation frequency is imposed, we used tonic rather than phasic stimulation, activating TRN using constant light for 30 s. Tonic TRN activation produced an immediate and substantial increase in low-frequency power in the local field potential (LFP) of ipsilateral somatosensory cortex (*Figure 1b–d*). This power increase was specific to the delta (1-–4 Hz) band, which increased by 2.56 dB (95% confidence interval [CI] = [2.13 2.97]) during laser stimulation. In contrast, beta and gamma (15–50 Hz) power decreased slightly (*Figure 1c*, median = -1.03 dB, CI = [-1.24 -0.84]). The increase in delta power was rapid and robust: delta waves were already evident in the first second of TRN activation (change = 1.12 dB, CI = [0.48 1.76]) and persisted throughout the stimulation period (*Figure 1b*). When individual slow wave events were detected automatically by thresholding filtered LFPs (see Materials and methods), 0.3 slow waves per second were detected during TRN stimulation, significantly more than baseline (increase = 0.13 events/s, CI = [0.10 0.17]). The amplitude of the negative-going peak was smaller during stimulation (change = -148 µV, CI = [-244 -54]), whereas the amplitude of the positive-going peak was larger during stimulation (change = 43 µV, CI = [8.6 76.7]), similar to the asymmetric waveforms typically seen during sleep slow waves (*Vyazovskiy et al., 2009*). The precise frequency and amplitude of slow wave events depend on the detection criteria being used, but these statistics nevertheless indicate a substantial increase in slow waves during TRN stimulation. To test whether this effect was TRN-specific rather than due to long-range GABAergic projections to thalamus, we next studied VGAT-Cre mice injected with AAV-EF1a-DIO-ChR2-EYFP specifically into the TRN and replicated the increase in delta (*Figure 1—figure supplement 3–5*). No such effect was observed in littermate mice that were negative for ChR2 (*Figure 1—figure supplement 6*), indicating that the slow waves were not due to nonspecific light or heating effects.

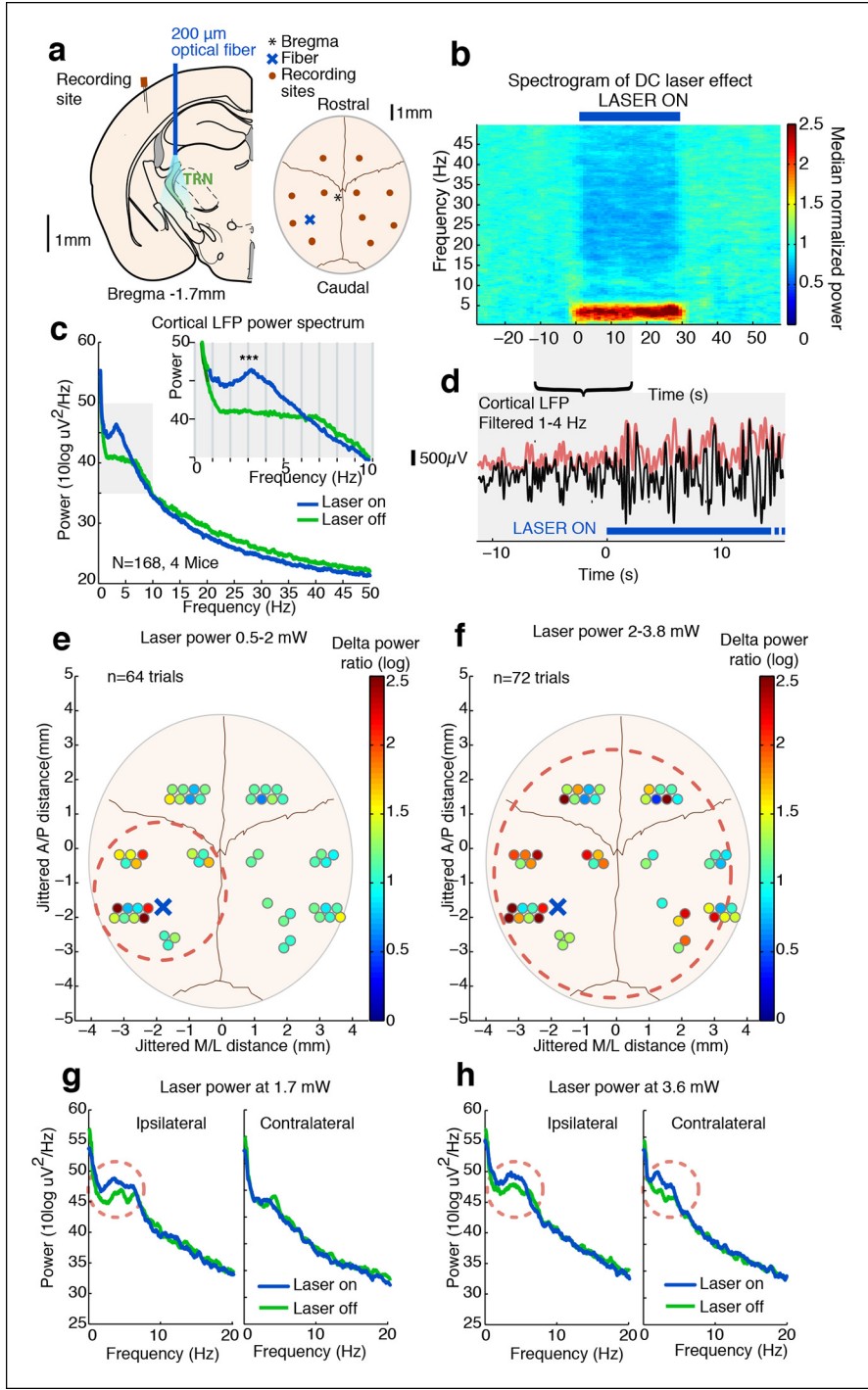

**Figure 1.** Tonic optogenetic stimulation of thalamic reticular neurons produces local cortical slow waves. (**a**) Diagram of surgery: fiber is implanted into left TRN, and stereotrodes are implanted in multiple sites across cortex. (**b**) Spectrogram showing average effect in ipsilateral somatosensory cortex across 168 trials (4 mice): TRN stimulation causes a rapid increase in delta (1–4 Hz) power that persists throughout the stimulation period. Power is normalized to the 30 s pre-stimulus period. (**c**) Average spectrum of LFP in somatosensory cortex: during tonic optogenetic activation of TRN, this cortical site demonstrates an increase in delta (1–4 Hz) power and a decrease in beta and gamma (12–50 Hz) power. Gray region shows zoomed-in plot of delta power increase. (**d**) Example trace from a single trial, showing the LFP filtered between 1–4 Hz (black line), and the instantaneous delta amplitude (red line). (**e** and **f**) Circles represent single electrodes, and their color indicates the size of the delta (1–4 Hz) power increase when laser is on (total n = 136 trials, 4 mice, 12–14 electrodes per mouse). At low powers (<2

*Figure 1. continued on next page*

*Figure 1. Continued*

mW), slow waves are induced only in electrodes near ipsilateral somatosensory cortex (red dashed circle). At high powers (>2 mW) that activate larger regions of TRN, slow waves appear across multiple cortical areas, including frontal cortex and contralateral cortex (red dashed circle). Distances are jittered so that electrodes from all mice can be displayed in a single schematic. Blue 'X' indicates placement of laser fiber. (**g**) Example spectra from one mouse at low laser power in electrodes ipsilateral and contralateral to the laser fiber (n = 10 trials): slow waves are induced in ipsilateral cortex but not in contralateral cortex. (**h**) Example spectra from same mouse at high laser power (n = 9 trials): slow waves are generated in both ipsilateral and contralateral cortex.

The following figure supplements are available for Figure 1:

**Figure supplement 1.** Selective TRN stimulation causes the induction of cortical slow waves.

**Figure supplement 2.** Example of VGAT-ChR2 mouse histology at 10x.

**Figure supplement 3.** Spectra of cortical LFPs recorded in VGAT-Cre mice expressing ChR2 selectively in TRN through local injections.

**Figure supplement 4.** Normalized spectrogram recorded in VGAT-Cre mice expressing ChR2 selectively in TRN through local injections.

**Figure supplement 5.** After viral injections, ChR2 expresses selectively in TRN.

**Figure supplement 6.** Slow wave induction depends on ChR2 expression.

**Figure supplement 7.** Simulation of light transmission through tissue at different laser powers.

**Figure supplement 8.** Phase offsets across cortex during TRN stimulation.

**Figure supplement 9.** Phase offsets across cortex are not correlated with distance to the electrode.

## TRN activation selectively controls a local ipsilateral cortical region

Cortical slow waves are observed locally in awake sleep-deprived animals, and this 'local sleep' correlates with decreased performance on cognitive tasks (*Vyazovskiy et al., 2011*). Given that TRN establishes topographical connections with its cortical inputs and thalamic outputs, we hypothesized that TRN could support local slow wave generation. We manipulated the extent of TRN activation by varying laser power, stimulating at low (<2 mW) or high (>2 mW) power while simultaneously recording local field potentials across cortex in individual mice (*Figure 1a*) to investigate the spatial spread of induced slow waves. We recorded in four awake head-fixed mice with fibers targeting the somatosensory sector of TRN. This stimulation protocol is expected to stimulate local somatosensory TRN at low laser power, and stimulate broader regions of TRN at higher laser power (*Figure 1—figure supplement 7*, (*Yizhar et al., 2011*)). We found that low laser power consistently enhanced local delta power in ipsilateral S1 (*Figure 1e*). Across all electrodes in the ipsilateral posterior quadrant (*Figure 1e*, red circle), 9/20 recording sites (45%) showed a significant increase in delta power during tonic activation (p<0.05, signed-rank test with Bonferroni correction). In contrast, only 2/32 recording sites (6%) in other cortical regions (e.g. contralateral or frontal) showed a significant increase in delta power, a significantly lower proportion than in the ipsilateral posterior quadrant (diff. = 0.37, CI = [0.15 0.6], binomial bootstrap [see Materials and methods]), demonstrating that slow waves were selectively induced in a local ipsilateral cortical region (*Figure 1e,g*). Trials using high laser power (i.e. with light spreading to larger regions of TRN) induced slow waves across a large cortical area: 10/20 (50%) of electrodes in the associated cortex and 11/32 (34%) of distant electrodes showed a significant increase in delta power (*Figure 1f,h*). The proportion of distant electrodes showing increased delta power was significantly higher than in the low laser power condition (diff. = 0.26, CI = [0.08 0.44], binomial bootstrap [see Materials and methods]). These data suggest that weak tonic activation of a small population of TRN neurons produces slow waves in a local ipsilateral cortical region, and that the strength of TRN activation controls the spatial spread of cortical

slow waves. Local activation of TRN thus controls an aligned region of cortex, and could support the spatially restricted slow waves that occur in local sleep.

Global cortical slow waves could be caused by broad thalamic inhibition, or by traveling waves across cortex spreading from the local site. To investigate these possibilities, we analyzed the phase relationships between different cortical sites during global induction of slow wave activity. We selected all channels with a significant increase in delta power during the high laser power stimulation, filtered the LFP between 1-–4 Hz, and quantified each electrode's phase relationship relative to the electrode closest to the optical fiber using the phase-locking value (PLV, [*Lachaux et al., 1999*]). At baseline, all channels in both the ipsilateral posterior quadrant channels and the more distant channels were significantly phase-locked to the reference site (ipsilateral: mean PLV = 0.52, s.d. = 0.13; distant: mean PLV = 0.43, s.d. = 0.15). During TRN stimulation, the PLV decreased slightly across all electrodes (p=0.0065, signed rank test) but remained significantly larger than chance (ipsilateral: mean PLV = 0.48, s.d = 0.15; distant: mean PLV = 0.39, s.d. = 0.17, every channel significant in permutation test). The mean phase offset at baseline was 0.10 radians (std. = 0.22 rad), and did not change significantly during TRN stimulation (mean = 0.15 rad, std. = 0.26 rad, p=0.45, signed rank test). The mean phase offsets across channels were clustered around zero, indicating that the induced slow wave activity was generally synchronized across cortex and phase lags were relatively short. Within individual electrodes, the phase lag was strongly correlated across the baseline and stimulation conditions (R = 0.93, CI = [0.82 0.98], *Figure 1—figure supplement 8–9*). These results suggest that TRN stimulation did not strongly affect cortical synchronization, but rather that the induced slow waves had similar phase relationships to the baseline dynamics across cortical regions. This result is consistent with previous findings that anesthesia-induced slow waves exhibit similar phase offsets to awake cortical dynamics (*Lewis et al., 2012*). This evidence supports the idea that TRN can induce slow waves in local or global cortical regions, with different thalamocortical loops supporting oscillations with different phase offsets across cortical sites.

## Cortical units rapidly phase-lock to induced slow waves and undergo OFF periods

Cortical slow waves during local sleep and NREM sleep mark an alternation of cortical spiking between activated (ON) and inactivated (OFF) states (*Vyazovskiy et al., 2009*; *Steriade et al., 1993*). To investigate whether the TRN-induced slow waves reproduced this pattern, we identified 31 single units (putative single neurons) across cortex and tested whether they were modulated by local slow waves. While TRN stimulation did not significantly change firing rates in cortical units (*Figure 2a*, median = -0.05 Hz, CI = [-0.17 0.09]), units from electrodes with induced slow waves became strongly phase-locked, similar to cortical activity during NREM sleep and local sleep (*Figure 2b,c*, median change = 0.044 bits, CI = [0.0012 0.112], n = 13 units). High gamma (70-–100 Hz) power, which correlates with multi-unit spiking (*Ray and Maunsell 1993*), also rapidly became phase-locked to slow waves during TRN activation (*Figure 2d*, median = 0.0015, CI = [0.0009 0.0022]), indicating that local neuronal activity was broadly locked to the induced slow waves. In contrast, increased phase-locking was not observed in units from electrodes with no induced slow waves (*Figure 2b*, median = 0.0002 bits, CI = [-0.004 0.007], n = 18 units). Across all units, the increase in phase-locking was correlated with the increase in LFP delta power (R = 0.77, CI = [0.57 0.88]).

Phase-locking analysis does not explicitly investigate OFF states, so we next automatically detected OFF states in electrodes that contained both MUA activity and TRN-induced slow waves (*Figure 2f*). During TRN activation, cortical neurons spent 13.1% of the time in OFF periods, significantly more than in the awake baseline state (7.04%, p<0.001 in each mouse) and significantly more than would be expected to occur randomly (3.09%, CI = [0.04 6.15]). In addition, these OFF periods occurred predominantly during the negative deflection of the slow wave (p<0.001 in each mouse, Pearson's chi-square test, *Figure 2e*). Their average duration was 122 ms (quartiles = [69 146] ms), and their mean frequency was 0.998/second, similar to natural sleep (*Vyazovskiy et al., 2009*). We concluded that the induced slow waves are sleeplike, marking an oscillatory pattern in which cortical neurons undergo periods of silence lasting tens or hundreds of milliseconds.

To determine the timescale of the shift into sleeplike dynamics, we computed the mean LFP and spike rate locked to laser onset, across all cortical units with local slow waves. The LFP underwent a negative-going deflection for the first 100 ms of laser stimulation (*Figure 2g*, top), and spike rates significantly decreased (*Figure 2g*, bottom). The reduction in cortical spiking was significant within

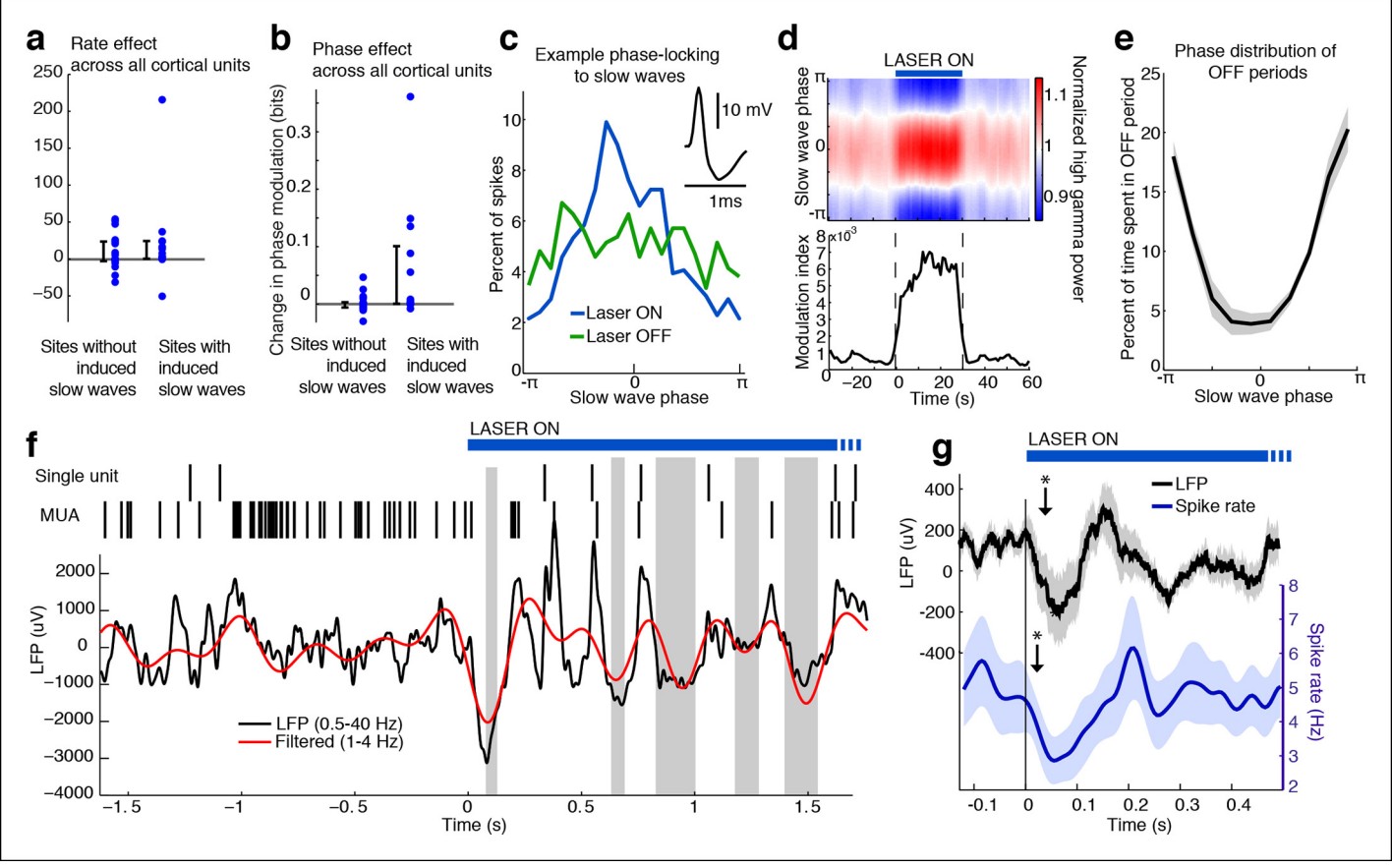

**Figure 2.** Cortical units undergo OFF periods that are phase-locked to the slow waves during TRN activation. (**a**) Rate effect across all cortical units, categorized by strength of delta power increase in that channel. There is no significant change in spike rate for either group. Error bars show interquartile range. (**b**) Phase-locking effects across all cortical units show that units on channels with induced slow waves become phase-locked to the slow waves during TRN stimulation. Error bars show interquartile range. Each dot is one unit (sites with slow waves: 13 units, 4 mice; sites without slow waves: 18 units, 4 mice). (**c**) Phase distribution of spikes from an example cortical unit recorded on a channel with a 3.4 dB delta power increase during TRN activation: unit becomes phase-locked to the slow wave. (**d**) Phase distribution of normalized high gamma (70–100 Hz) power shows that high gamma power becomes rapidly phase-locked to slow waves during TRN stimulation. Gamma power is normalized to have a mean of 1 at each time point, so brightness indicates the strength of phase-locking. (**e**) Phase distribution of all OFF periods shows that they occur during the trough of the slow waves. (**f**) Example trace from somatosensory cortex: optogenetic TRN stimulation rapidly induces slow waves that are associated with OFF periods in cortical activity (gray shaded regions mark automatically detected OFF periods). (**g**) Mean spike rate and LFP locked to laser onset in channels with induced delta: the induced slow wave trough and phase-locked cortical inhibition are observed within 100 ms of laser onset. Stars indicate timing of significant ($\alpha = 0.05$) decrease in LFP voltage and mean spike rate; the decrease persists throughout the first 100 ms. Triggered LFP and units are averaged across cortical electrodes with a delta power increase (n = 14 channels, 4 mice), shaded region is std. err.

20 ms, and the LFP effect by 35 ms. The effect of TRN activation was therefore rapid, driving a slow wave and cortical suppression in tens of milliseconds, and thereby inducing an abrupt transition into a new cortical state in which neurons undergo rhythmic OFF periods.

## Stimulation of TRN causes suppression and rhythmic firing in thalamus

Tonic TRN stimulation produced striking cortical effects that were locally defined, suggesting that the key circuit mechanism was through thalamus, which is the main target of TRN outputs and has corticotopic projections that could support local control of cortex. To investigate the impact of TRN stimulation on thalamic activity, we recorded from TRN and nearby thalamus (targeting the ventral posteromedial nucleus). We isolated 28 single units from five mice and used spike waveforms to distinguish between putative TRN ('Narrow') and putative thalamocortical (TC, 'Wide') neurons (*Figure 3a*). The waveform distribution was bimodal (*Figure 3a*), with 'Narrow' units (peak-to-trough time under 200 μs), and 'Wide' units (peak-to-trough time above 200 μs). Narrow waveforms are

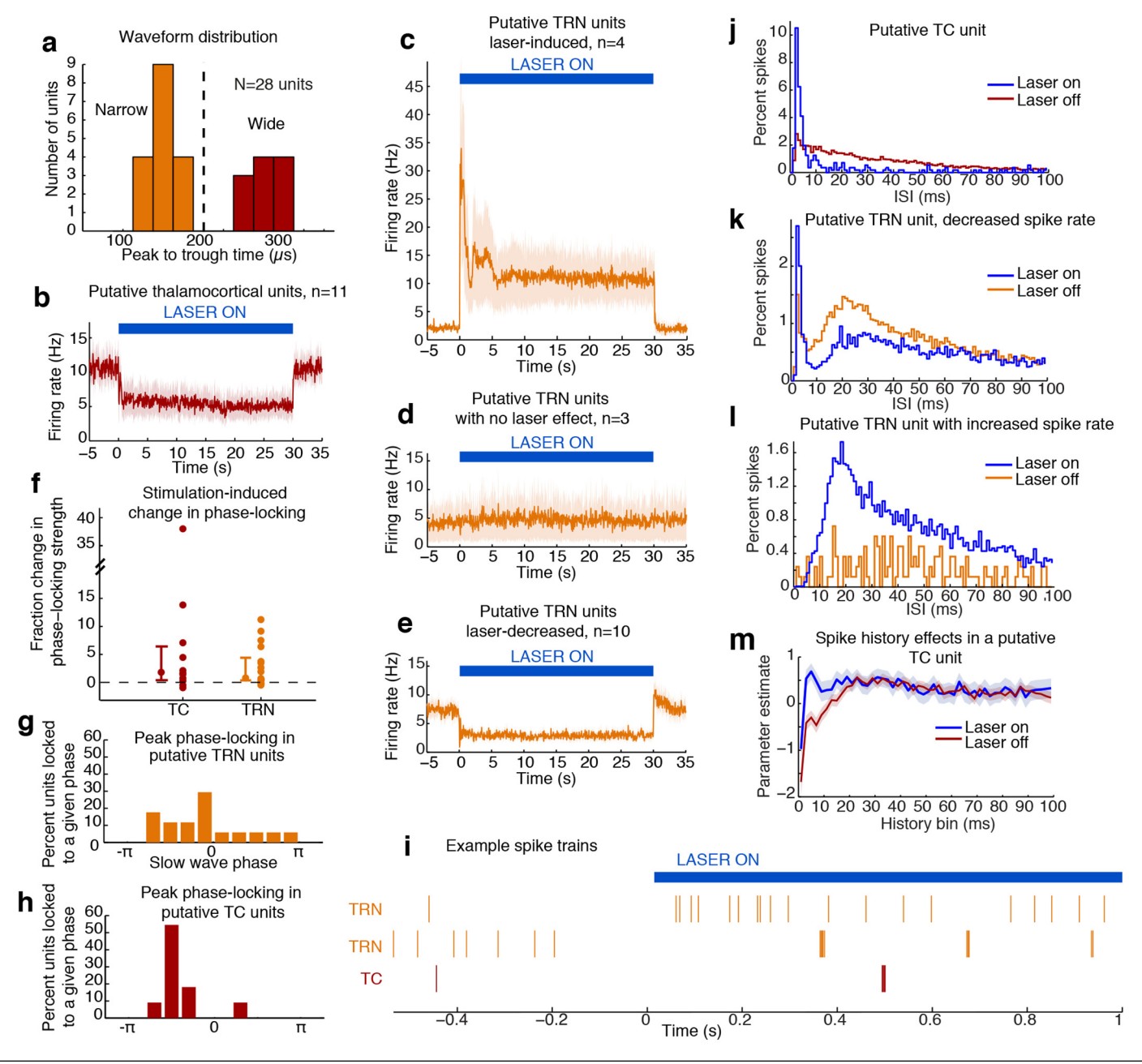

**Figure 3.** Optical stimulation strongly activates a subset of TRN neurons and induces periodic suppression of thalamic firing. (a) Histogram of waveform parameters from single units recorded in freely behaving mice show a bimodal distribution of peak-to-trough time across subcortical units (n = 28 units, 5 mice). Units with peak-to-trough times under 200 μs were categorized as Narrow (putative TRN), and units over 200 μs were categorized as Wide (putative thalamocortical [TC]). (b) Putative thalamocortical (Wide) units consistently decrease their firing rates during laser stimulation. Mean firing rate in 500 ms bins, shaded region is std. err. across units. (c–e) Heterogeneous firing rates in TRN during stimulation: 4 units strongly increase their firing rates, whereas 10 units decrease their firing rates. The modulation in firing rate is strongly time-locked to laser onset and offset. Shaded regions are std. err. across units. (f) Phase-locking effects across all subcortical units show that most become phase-locked to the slow waves during TRN stimulation. Circles mark the change in phase-locking for each unit; error bars show median change with 25th and 75th quartiles. (g) The phase distribution of putative TRN neurons is broad, with different neurons exhibiting different preferred phases. (h) Peak phase-locking values of putative TC neurons show a tight distribution (Kurtosis = 3.99, n = 11 units), indicating that nearly all putative TC neurons show similar phase-locking to the LFP. Putative TC phase-locking is more consistent across units than putative TRN phase-locking (in b; Kurtosis = 0.24, n = 17 units, group difference = 3.74, significant at α = 0.05 from bootstrap resampling). (i) Example spike rasters around laser onset from 3 single units. Units were not recorded simultaneously; each raster is an independent example. (j–l) Example ISI histograms in single units. (m) Example of parameter estimates from generalized linear model for one unit shows the contribution of recent (<10 ms) spike history increases during stimulation. Shaded regions are std. err.

*Figure 3. continued on next page*

*Figure 3. Continued*

The following figure supplements are available for Figure 3:

**Figure supplement 1.** Example waveforms for putative TC and TRN neurons.

typically characteristic of TRN GABA-ergic fast-spiking inhibitory neurons (*Zhao et al., 2011*; *Yizhar et al., 2011*), so we used these units (n = 17) to infer the activity of TRN, and the 'Wide' units to infer TC neuron activity (*Wang et al., 2010*). Spike rates in nearly all putative TC units (9/11, 81.8%) significantly decreased during laser stimulation, whereas no units significantly increased their spike rate (*Figure 3b*). Thalamic activity was therefore consistently suppressed by TRN activation.

Putative TRN units exhibited heterogeneous changes in firing rates, as expected due to the local stimulation induced by limited light spread (*Figure 3c–e*). A subset of units increased their spike rates as predicted (4/17 units, 23.5%), while other units had no significant change (17.7%), or decreased their firing rate significantly (58.8%). The magnitude of the increase in firing rates (*Figure 3c*) suggested that stimulation induced strong local excitation of TRN, and possibly led to downstream inhibition of neurons located farther from the optical fiber through either intra-TRN inhibition or through suppression of thalamic drive to TRN. Alternatively, the heterogeneous effects of stimulation on TRN neurons could reflect variable expression levels or heterogeneous cell types within the TRN with different functional properties (*Halassa et al., 2014*; *Barrionuevo et al., 1981*; *Lee et al., 2014*), leading to increased spike rates in a specific subset of TRN cells due to their cell type and role in the local circuit structure. In either scenario, high firing rates in a subset of TRN neurons is sufficient to consistently inhibit thalamocortical cells. These results demonstrate that optogenetic stimulation of TRN strongly drives only a local subpopulation of TRN neurons, but nevertheless causes consistent inhibition of thalamic activity.

Whether and how thalamic inhibition can generate sleep states has been debated: although thalamic activation induces wake states (*Poulet et al., 2012*), lesioning thalamus does not produce sleep states (*Constantinople and Bruno, 2011*). Similarly, while direct inhibition of thalamus does not induce slow waves (*Lemieux et al., 2014*), activating inhibitory brainstem projections to thalamus does (*Giber et al., 2015*), and thalamic stimulation can entrain cortical slow waves (*David et al., 2013*). We therefore hypothesized that thalamus participates in generating slow waves, and tested whether these neurons engaged in the induced rhythm. We indeed observed that subcortical units increased their phase-locking to the thalamic LFP slow waves (*Figure 3f*). Putative TRN unit phase-locking was diverse: laser stimulation increased overall phase-locking (*Figure 3c*, increase = 0.027 bits, CI = [0.005 0.041]), but the preferred phase varied substantially across units (*Figure 3g*). In contrast, putative TC neurons consistently increased their phase-locking during stimulation (*Figure 3f,h*, increase = 0.011 bits, CI = [0.001 0.022]). Stimulation of TRN thus causes thalamic neurons to oscillate in a slow wave pattern rather than undergoing a simple decrease in activity.

Thalamic entrainment to slow waves can be a combination of intrinsic mechanisms (*Wang, 1994*; *McCormick and Pape, 1990*) and cortical entrainment *Steriade et al., 1991* To investigate the contribution of intrinsic oscillatory mechanisms, we examined thalamic spike properties, as thalamic cells can burst at delta frequencies during hyperpolarization (*McCormick and Pape, 1990*). To test whether our manipulation affected bursting, we fit generalized linear models to the spike trains of each unit. We tested whether spike history 2-–4 ms prior predicted an increased likelihood of spiking during TRN stimulation as compared to baseline (*Figure 3m*). We found a significant change in 7/11 (63%) of putative TC cells, suggesting that TRN stimulation increased the likelihood of thalamic bursting (*Figure 3j,m*). In addition, 5/10 of the TRN units that decreased firing rates during stimulation increased their 2–4 ms history dependence, whereas 0/4 of the TRN units with increased firing did. These results suggest that the laser-driven TRN units fire tonically (*Figure 3i,l*), and lead to bursting and phase-locking in neighbouring TRN cells and in thalamus (*Figure 3i–k*).

## Tonic TRN activation decreases behavioural arousal state

Slow wave activity is associated with drowsiness and sleep (*Pace-Schott and Hobson, 2002*), and thalamic activity plays an important role in awake states (*Alkire et al., 2000*; *Schiff, 2008*), so we next investigated whether strong TRN activation produced behavioral signs of decreased arousal, by

recording electromyography (EMG) and frontal electroencephalography (EEG) in freely behaving mice. EMG power decreased significantly during TRN stimulation (*Figure 4a*, mean = -0.06, CI = [-0.08 -0.04]), indicating that stimulation caused the animals to become less active. The decrease was significant within 1 s of laser onset, demonstrating rapid modulation of behavioral state. In addition, the EEG and EMG effects were significantly negatively correlated on the single trial level (*Figure 4c*, correlation coefficient = -0.43, CI = [-0.53 -0.33]). This correlation was significantly stronger than at randomly shuffled times, (p < 0.05, bootstrap) demonstrating that the decrease in arousal was specifically associated with the optogenetically induced slow waves. Control experiments in ChR2 negative littermates showed no EMG effect (*Figure 4—figure supplement 1*). To test a more general measure of arousal, we recorded videos of behaving mice and used an automatic video scoring algorithm to quantify their motion. Motion decreased significantly during TRN activation (*Figure 4a*, decrease in 58.0% of trials, CI = [53.1 62.7]). These results demonstrated that TRN activation causes a rapid decrease in arousal state, evident by a decline in motor activity.

We next investigated whether the behavioral effect was due to a decrease in motion during the awake state, or whether the mice were also sleeping more during TRN activation. We performed semi-automated sleep scoring using EMG and frontal EEG recordings and found that TRN stimulation reduced awake time (median = -2.1 percentage points, CI = [-4.2 -0.47]) and increased NREM sleep (median = 3.5 percentage points, CI = [1.68 5.44]) (*Figure 4b*). Tonic TRN activation thus shifted sleep dynamics, biasing animals towards NREM sleep. The change in behavioural state was subtle, corresponding to a decrease in motor activity and a small increase in the probability of NREM sleep, similar to the awake but drowsy behaviour reported during local sleep.

## Partial inhibition of TRN decreases slow wave activity during sleep

Given that stimulating TRN could rapidly and locally induce cortical slow waves, we asked whether inhibiting TRN in a sleeping animal could reduce its cortical slow wave activity. We expressed halorhodopsin in TRN neurons using local viral injections (*Figure 4—figure supplement 2*). resulting in widespread expression within a local region of TRN (*Figure 4—figure supplement 3*), and recorded cortical LFPs during partial TRN inhibition. We found that TRN inhibition reduced slow waves in mice during NREM sleep (change = -0.45 dB, CI = [-0.77,-0.13], *Figure 4d*). To ensure that this effect was not due to spontaneous awakenings, we shuffled the laser onset times and did not observe any effect (shuffled change = 0.01, CI = [-0.39, 0.32]), suggesting that the decrease in slow wave activity was specifically due to TRN inhibition. We did not observe behavioural effects of TRN inhibition (e. g., *Video 1*), which likely reflects that multiple powerful pathways including brainstem are acting to suppress motor activity during NREM sleep rather than TRN alone (*Lydic and Baghdoyan, 2005*), but could also indicate that more extensive suppression of TRN is needed to modulate behaviour than can be achieved in this preparation. We therefore concluded that TRN can bidirectionally modulate cortical slow wave activity.

## Tonic TRN stimulation further increases slow wave activity during sleep

For direct comparison to the halorhodopsin experiments, we also tonically stimulated TRN in VGAT-Cre mice expressing ChR2 during natural NREM sleep. Tonic TRN stimulation applied during NREM sleep increased delta power by 0.62 dB (CI = [0.27 0.97], *Figure 4d*), indicating a further induction of cortical slow waves even during sleep states when slow waves are already present. In contrast, spindle (9–15 Hz) power decreased significantly (median = -0.80 dB, CI = [-1.17 -0.42]), likely due to increased number and prolongation of OFF periods. These dynamics are similar to those observed during transitions into deeper stages of sleep, as spindles subside and slow waves increase, suggesting that TRN stimulation can shift cortical dynamics into deeper stages of NREM.

## TRN stimulation modulates existing slow waves during anesthesia

We next examined whether TRN can also decrease arousal in anesthetized mice; would tonic TRN activation induce slow waves when the animal is already in a state of decreased arousal and exhibits global slow waves? We recorded EEG during isoflurane anesthesia and found that the baseline delta power was high, and there was no further increase during TRN activation (*Figure 4e*), suggesting that the ability of TRN to generate slow waves was saturated. When individual slow wave events were detected, they also showed no change in frequency during TRN stimulation (baseline = 0.30

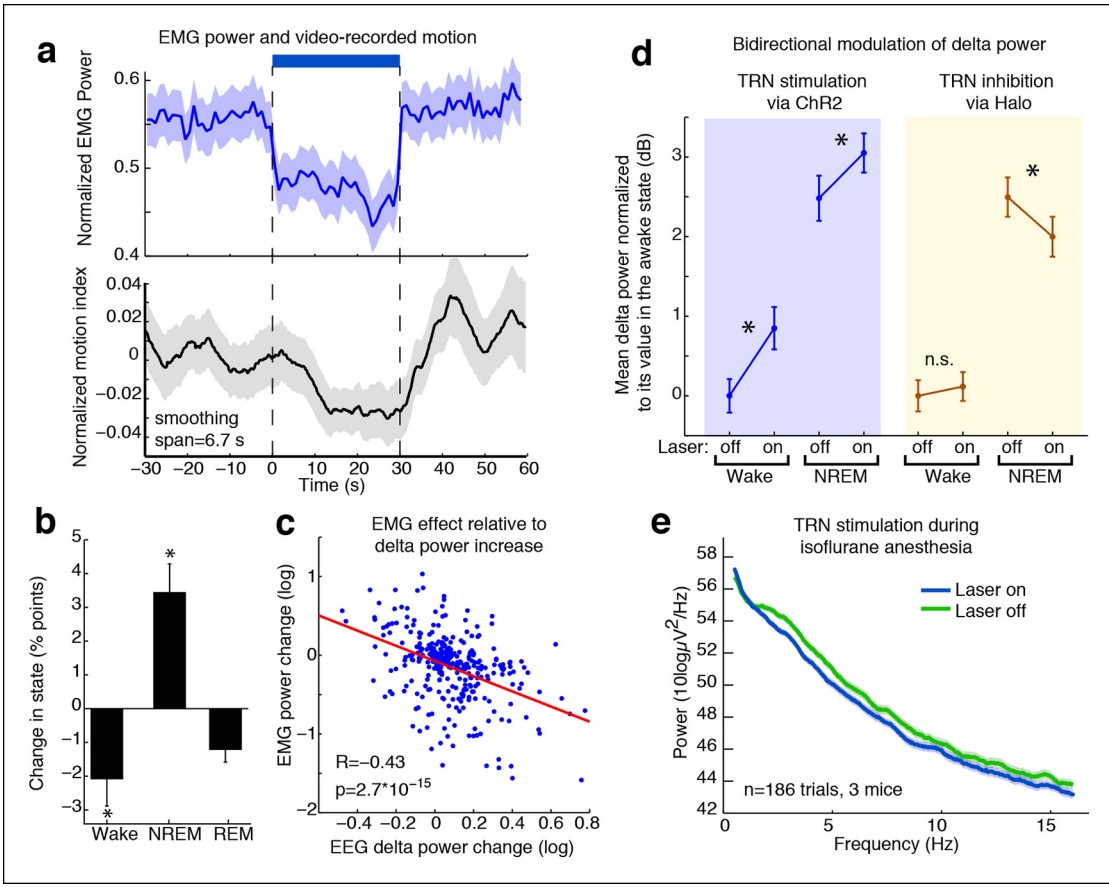

**Figure 4.** TRN modulates arousal state in a bidirectional and state-dependent manner. (a) Top panel: Mean EMG power locked to laser onset shows that EMG power decreases significantly during unilateral TRN stimulation in freely behaving mice (n = 315 trials, 8 sessions, 2 mice). Bottom panel: Mean smoothed motion (6.67 s moving average) detected in video: animals' motion decreases significantly during optogenetic stimulation (n = 421 trials, 7 mice). (b) Mean change in arousal state during TRN activation: mice spend significantly more time in non-REM sleep and significantly less time in the awake state (n = 560 trials, 3 mice). Stars indicate significant effects at α = 0.05. (c) Individual trial correlation shows that the decrease in EMG power is correlated with the TRN-induced increase in EEG delta power (n = 315 trials, 2 mice). (d) Delta power increases in VGAT-Cre mice expressing ChR2 during TRN stimulation, whether awake or in NREM at time of stimulation. In VGAT-Cre mice expressing halorhodopsin, TRN inhibition has no effect in awake mice, whereas it decreases the delta power that is present in sleeping mice. N = 3 mice expressing ChR2 (160 wake trials; 192 NREM trials), n = 3 mice expressing Halo (459 wake trials; 211 NREM trials), stim. duration = 5 s. All recordings were in freely behaving mice. Dots show mean power +/- std. err; stars indicate a significant effect of the laser on the median power, computed with the Wilcoxon signed-rank test. (e) Cortical recordings in VGAT-ChR2 mice (n = 186 trials, 3 mice). During isoflurane anesthesia, the slow waves appear to be saturated and are not increased by TRN stimulation. Instead, broadband power decreases, suggesting a shift in dynamics that favours the inactivated state.

The following figure supplements are available for Figure 4:

**Figure supplement 1.** Laser-induced behavioural decreases in arousal depend on ChR2 expression.

**Figure supplement 2.** Halorhodopsin expresses in TRN.

**Figure supplement 3.** Halorhodopsin expresses in most cell bodies within the locally injected region of TRN, and not in thalamic cell bodies outside TRN.

**Figure supplement 4.** TRN stimulation further increases cortical neuronal phase modulation during isoflurane anesthesia.

*Figure 4. Continued*

**Figure supplement 5.** Example of state-dependent increases in cortical phase-locking during isoflurane anesthesia.

**Figure supplement 6.** TRN stimulation deepens thalamic neuronal suppression during isoflurane anesthesia.

events/s, stimulated = 0.29 events/s). Instead the EEG showed a broadband (0.5-–50 Hz) decrease in power (-0.53 dB, CI = [-0.69 -0.37]), demonstrating a generalized quieting of cortical activity. Furthermore, the fraction of time spent in OFF periods increased by 4.02 percentage points (CI = [1.9 6.2]) and the amplitude of the positive-going LFP slow wave peak increased by 45 µV (CI = [16 75]) Cortical units increased their phase-locking to slow waves (*Figure 4—figure supplement 4–5*, median = 0.06 bits, CI = [0.018 0.189]), while their firing rates decreased (median = -0.09 Hz [-5.5%], CI = [-0.22 -0.02]), suggesting that cortical activity became more strongly suppressed by the existing slow waves. Similarly, firing rates in putative thalamocortical neurons were suppressed to even lower levels by TRN stimulation during anesthesia (*Figure 4—figure supplement 6*). We concluded that the anesthetized cortex is shifted into an even deeper state by TRN activation: not by inducing slow waves, but rather by modulating the dynamics of a slow wave that is already present and thereby prolonging the duration of the periodic suppressions.

## Discussion

States of decreased arousal are marked by local cortical slow waves, but the circuit mechanisms that induce these states and their causal link to arousal are unknown. In this study, we identified a local thalamocortical circuit that modulates cortical arousal state. Specifically, we found that tonic TRN activation mediates an increase in thalamic inhibition and produces sleep-like cortical slow waves whose spatial spread depends on the extent of TRN activation. This electrophysiological effect is correlated with an optogenetically-induced reduction in behavioral arousal. TRN-mediated thalamic inhibition can thus serve as a mechanism for local modulation of cortical arousal state.

### Slow waves are generated by local corticotopic circuits

We find that TRN can selectively induce slow waves in local cortical regions. This result reinforces recent findings suggesting that sleep contains dynamics that are differentiated across cortex rather than a globally homogeneous cortical state (*Krueger et al., 2008*). Awake sleep-deprived rats also exhibit slow waves and OFF periods in local cortical areas (*Vyazovskiy et al., 2011*), and these local dynamics are correlated with behavioural deficits. Our results show that localized depolarization in TRN can produce such local oscillations, and could therefore underlie the fragmented cortical slow waves observed during sleep as well as drowsy awake states. In addition, local cortical OFF states have been observed during sleep (*Nir et al., 2011*) and general anesthesia (*Lewis et al., 2012*) in human subjects, demonstrating that OFF periods frequently occur locally even when slow-wave activity is present throughout cortex. The observed asynchronous slow waves in these unconscious states could be due to a global activation of TRN, producing slow waves throughout cortex, but different cortical regions are associated with specific thalamocortical circuits that enable them to undergo separate and asynchronous oscillations. Finally, the local control that TRN exerts over cortex provides evidence for how TRN could modulate attention across sensory modalities, by suppressing arousal in specific cortical regions. This finding thus supports the theory that TRN could function to modulate attention, not only by gating thalamic transmission of sensory information to cortex (*Crick, 1984*), but also by modulating non-sensory-driven thalamic activity, which controls the ongoing state in local cortical regions and thereby influences the structure of functional networks in cortex. The finding that TRN can independently control limited corticothalamic circuits therefore suggests it could serve as a central circuit mechanism to regulate specific cortical regions, modulating both attention and arousal.

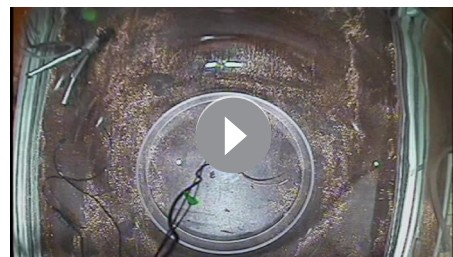

**Video 1.** Laser stimulation in a mouse expressing NpHR during NREM sleep. The mouse maintains NREM sleep during the stimulation and the light alone does not cause visible behavioural changes.

## TRN-induced thalamocortical slow waves show rapid onset

In natural behavior, animals can rapidly transition between arousal states. We find that the slow waves induced by depolarization of TRN are initiated abruptly, suggesting it could play a role in rapid state modulation. Cortical activity is suppressed within 20 ms of laser onset, and the deflection in the LFP can be detected within 35 ms. This pattern suggests that tens of milliseconds of TRN activation are sufficient to inhibit thalamic input to cortex and produce a cortical OFF state. The dynamics at laser offset are similarly abrupt, with slow waves vanishing within a second. TRN can therefore serve as a rapid modulator of arousal state. This finding is also compatible with established neuromodulatory sleep circuits (*Pace-Schott and Hobson, 2002*), such as monoaminergic arousal pathways (*Saper et al., 2005*), as these neuromodulators affect TRN activity as one component of arousal regulation. TRN thus engages a fast-acting circuit for arousal control, demonstrating that thalamocortical loops can rapidly control cortical arousal state.

## TRN supports both slow waves and sleep spindles

Here we used tonic and low-power activation of TRN, leading to decreased thalamic firing rates without complete suppression. This tonic paradigm induced slow waves and modulated cortical state in awake mice without affecting power in the spindle band (7–15 Hz). In sleeping mice, tonic activation further increased slow waves and decreased spindle power, similar to the dynamics that occur during transitions into deeper stages of sleep. Interestingly, strong phasic activation of TRN induces spindles during non-REM sleep but not in the awake state (*Halassa et al., 2011*; *Barthó et al., 2014*). Phasic and tonic modulation of TRN activity therefore produce qualitatively different sleep oscillations, suggesting that changes in the dynamics of inputs to TRN could underlie shifts between different stages of sleep. Brief activation of TRN may lead to a thalamic burst that entrains a cortical spindle, whereas prolonged activation hyperpolarizes thalamocortical cells and allows intrinsic slow waves to emerge. Interestingly, a recent study demonstrated that high-frequency vs. low-frequency corticothalamic input produces qualitatively different effects in thalamus (*Crandall et al., 2015*), consistent with the idea that modulating the temporal dynamics of input to the thalamocortical circuit can lead to different arousal states.

## Circuit mechanisms underlying the induced thalamocortical slow waves

The circuit mechanism that generates slow wave activity during sleep and anesthesia remains a topic of debate (*McCormick and Bal, 1997*; *Crunelli and Hughes, 2010*; *Destexhe and Contreras, 2011*). Our results show that slow waves can be produced by depolarizing TRN, and suggest they may be generated through overall inhibition of thalamic input to cortex. Our manipulation activated a local population of TRN neurons that inhibit the associated region of thalamus. In the absence of this thalamic input to cortex, which drives the desynchronized cortical state (*Poulet et al., 2012*), cortex and thalamus jointly enter an oscillation in which activity is periodically suppressed. Previous studies have demonstrated that cortex can maintain an awake state even when thalamus is lesioned or inactivated (*Constantinople and Bruno, 2011*; *Zagha et al., 2013*); our results therefore suggest that slow waves in the intact brain require the involvement of cortical, TRN, and TC neurons in a coordinated rhythm. Direct inhibition or lesioning of TC cells may disrupt the coordination of this rhythm, whereas physiological levels of inhibition from TRN may allow the emergence of intrinsic thalamic delta. Furthermore, TRN stimulation did not induce slow waves in the anesthetized animal, when thalamic T-type calcium channels are blocked (*Todorovic and Lingle, 1998*), suggesting that thalamus may contribute to slow wave generation. This theory is also consistent with studies showing that anesthetic infusions directly into thalamus can induce slow waves (*Zhang et al., 2012*), and that manipulations of thalamus affect the frequency of slow waves (*David et al., 2013*). Taken together,

these findings suggest that TRN-mediated inhibition of thalamus is a robust driver of local slow wave activity, and that slow waves in the intact brain may require both cortical and TC neurons to fire in a coordinated rhythm. Hyperpolarization but not complete suppression of thalamus may be key to generating slow waves, as TRN induces partial suppression and bursting in TC neurons (*Figure 3b,j*), as opposed to the stronger suppression achieved through direct manipulation of thalamus. However, TRN, thalamus, and cortex are independently capable of generating low-frequency rhythms (*Amzica and Steriade, 1998*; *Zhang et al., 2009*; *Beltramo et al., 2013*; *Dossi et al., 1992*) and our results could be consistent with any or all of these areas acting as the slow wave pacemaker. Slow waves could arise through thalamic oscillations, could be generated in cortex due to withdrawal of thalamic excitatory drive, or could be jointly driven by both structures. In each scenario, TRN may act as a local regulator that can shift the thalamocortical circuit between desynchronized and oscillatory regimes.

## Neuronal activity in TRN across arousal states

Due to the technical challenges in recording from TRN, only a small number of previous studies have reported single unit recordings in TRN across arousal states. Interestingly, several reports have observed heterogeneous firing properties during sleep, and have suggested the possibility of multiple types of TRN neurons that play different roles in arousal state (*Halassa et al., 2014*; *Barrionuevo et al., 1981*). Such heterogeneity could explain the variable firing properties observed in different studies. While many TRN neurons decrease their firing rates during NREM sleep, a subset maintain or increase their firing rates (*Barrionuevo et al., 1981*; *Steriade et al., 1986*). In addition, most TRN neurons exhibit bursting properties during sleep, with brief periods of activity locked to slow wave rhythms (*Halassa et al., 2014*; *Barrionuevo et al., 1981*; *Steriade et al., 1986*; *Marks and Roffwarg, 1993*). These observations are consistent with our results, in which we observe heterogeneous firing rates in TRN, but nearly all units exhibit phase-locking to the induced slow waves during stimulation. It may be that during natural sleep, high firing rates in TRN inhibit thalamic activity and thereby induce slow waves, but those high rates are limited to only certain phases of the slow wave (rather than tonic continuous firing) due to synchronized delta-range input from thalamus and cortex. Experiments using closed-loop control to stimulate at specific phases of slow wave activity could explore whether tonic or phase-locked activity in TRN is most effective at driving cortical slow waves. It may also be that a specific subtype of TRN neuron induces slow wave activity, and that the microcircuitry of TRN enables this subtype to fire more while suppressing other TRN neurons during optogenetic stimulation. Future studies could also examine the effect of stimulation across multiple regions of TRN, as there are distinct subnetworks within TRN that may play different functional roles in regulating arousal (*Halassa et al., 2014*; *Lee et al., 2014*).

## TRN activation as a component of general anesthesia

The finding that TRN activation induces slow waves and decreases arousal could contribute to a subset of the effects of GABAergic drugs used for general anesthesia, such as propofol. In human subjects, propofol induces a large increase in low-frequency (0.1–4 Hz) power (*Murphy et al., 2011*), and this slow wave induction has been suggested as a potential mechanism for unconsciousness (*Lewis et al., 2012*; *Massimini et al., 2009*). Propofol is a GABA-A agonist (*O'Shea et al., 2000*), suggesting that it could increase low-frequency EEG power by increasing the inhibitory effects of both TRN and brainstem structures on thalamus (*Alkire et al., 2000*). Decreased thalamic activity has also been implicated in disorders of consciousness (*Lutkenhoff et al., 2013*), and may be a potent mechanism for inducing decreased arousal (*Schiff, 2008*). Modulation of thalamic activity may therefore be an important component of general anesthesia.

## TRN as one element of arousal control

Partial inhibition of TRN during NREM sleep caused a reduction in slow wave activity, suggesting that TRN plays a role in slow waves observed during natural sleep. However, the decrease in power was modest. This small effect size could be partially due to incomplete inhibition of TRN due to local light delivery and incomplete expression, as little is known about the structure of intra-TRN circuits, and inhibiting only a subset of TRN cells may have different effects than inhibiting all of them. However, the small effect size likely also reflects the fact that multiple arousal centers, including many

brainstem nuclei, are modulated during NREM sleep (*Pace-Schott and Hobson, 2002*; *Saper et al., 2010*), leading to broad thalamic inhibition. Suppressing TRN would thus only moderately reduce inhibitory input to the thalamus as other sources of inhibition persist, leading a reduction in slow wave activity rather than complete suppression. Similarly, stimulating TRN led to robust local cortical slow waves and a relatively small decrease in behavioural arousal, suggesting TRN activity drove local sleep and drowsiness more often than a complete transition into global sleep. Our results, in combination with previous studies, suggest that TRN acts as only one element of a redundant circuit for arousal control. Brainstem structures modulate global arousal state, whereas TRN may serve as a spatially selective circuit for fine-tuning arousal state across local cortical regions, allowing flexible modulation of slow wave activity. TRN may thus play a role in the local slow waves that subserve sleep-dependent memory consolidation, whereas brainstem would regulate the presence of sleep vs. wake states at a global scale.

### TRN controls local cortical arousal state

We conclude that TRN can selectively induce slow waves in local cortical regions. Taken together, our results demonstrate that TRN can control oscillatory dynamics in local thalamocortical circuits and suggest it could serve as a spatially selective circuit mechanism to rapidly and independently modulate cortical arousal.

## Materials and methods

### Optogenetic manipulation

All experimental procedures were approved by the MIT Committee on Animal Care. ChR2 expression was achieved through use of either viral injections targeted at TRN in VGAT-Cre mice (n = 3 mice, *Figure 1—figure supplement 3,4*, *Figure 3d*) or through expression in VGAT-ChR2 mice (n = 11 mice, all remaining figures). VGAT-ChR2 mice were obtained from Prof. Guoping Feng's laboratory and VGAT-Cre mice were obtained commercially (Jackson Laboratory, stock number 106962, *Slc32a1*). For viral injections, an AAV-EF1a-DIO-ChR2-EYFP virus was injected into two sites bilaterally (A/P 0.6 mm, M/L 1 mm, D 3.75/3.25 mm; A/P 1.58 mm, M/L 1.9 mm, D 3 mm). Halorhodopsin experiments were done through injections of AAV-EF1a-DIO-eNpHR3.0-EYFP into the same sites as described above, again using VGAT-Cre mice (n = 3 mice). These viruses were produced by the vector core at University of North Carolina, Chapel Hill, with titers around $10^{12}$ VG/ml. A volume of 100–200 nL per injection site was used. Viral injections were immediately followed by implant of electrodes and optical fibers as described below. Mice with viral injections were implanted at least 3 weeks prior to beginning experiments to allow time for viral expression to develop.

### Cortical implants

In order to deliver light to TRN, all VGAT-ChR2 mice were implanted with a 0.21 NA fiber of 200 micron diameter targeting left TRN (1.8 mm lateral, -0.8 to -1.7 mm posterior relative to bregma; 2.1 mm deep). VGAT-Cre mice received implants of 2 to 4 optical fibers to allow for simultaneous manipulation of two sites in TRN. Two types of electrode implant were performed: either a cortical implant with stereotrodes (*McNaughton et al., 1983*) distributed across different cortical sites; or a subcortical implant, with moveable stereotrodes targeted to TRN. Mice did not receive both implant types; an individual mouse in which units were recorded would receive either a cortical or subcortical implant. For the cortical implants, stereotrodes were made from pairs of 12.5 micron nichrome wire gold plated to ~300 kOhm (California Fine Wire, Grover Beach CA). Electrodes were attached to small sections of plastic tubing cut to defined depth offsets and inserted by hand in 11 recording sites distributed across the cortex (*Figure 1a*) at depths of 400, 500, 600, or 1300 microns, as defined by the length of the electrode extending from the plastic tubing. In one mouse with the cortical implant, subcortical recordings were also acquired simultaneously by gluing a stereotrode to the optical fiber, with the stereotrode extending 200 microns beyond the optical fiber. This allowed acquisition of single thalamocortical units. To calculate laser power within the brain, the laser power was first measured outside of the brain, and then this value was scaled to account for diminished power after passing through the fiber. For surgery, mice were anesthetized with 1% isoflurane and

individual holes were drilled for electrode and optical fiber insertion. Electrodes were inserted by hand and the optical fiber was placed using a stereotaxic arm.

## Subcortical implants

Hyperdrive bodies were designed in 3D CAD software (SolidWorks, Concord, MA) and stereolithographically printed in Accura 55 plastic (American Precision Prototyping, Tulsa, OK). Each hyperdrive was loaded with 6–8 individual, independently movable microdrives made of a titanium screw cemented to a 21-gauge cannula. Each microdrive was loaded with 1–3, 12.5 micron nichrome stereotrodes (California Fine Wire Company, Grover Beach, CA), which were pinned to a custom-designed electrode interface board (EIB) (Sunstone Circuits, Mulino, OR). Two EMG wires, two EEG wires and one ground wire (A-M systems, Carlsborg, WA), were also affixed to the EIB. An optical fiber targeting TRN (Doric Lenses, Quebec, Canada) was glued to the EIB. TRN targeting was achieved by guiding stereotrodes and optical fiber through a linear array (dimensions ~1.1 x× 1.8 mm) secured to the bottom of the hyperdrive by cyanoacrylate. For surgery, mice were anesthetized with 1% isoflurane and placed in a stereotaxic frame. For each animal, five stainless-steel screws were implanted in the skull to provide EEG contacts (a prefrontal site and a cerebellar reference), ground (cerebellar), and mechanical support for the hyperdrive. A craniotomy of size ~3 × 2 mm was drilled with a center coordinate of (M/L 2.5 mm, A/P -0.5 mm). The implant was attached to a custom-designed stereotaxic arm, rotated 15 degrees about the median and lowered to the craniotomy. Stereotrodes were lowered slightly at the time of implantation (<500 microns) and implanted into the brain.

## Data acquisition

Electrophysiology was performed in a total of eight VGAT-ChR2 positive mice, six VGAT-Cre mice with viral injections, and three mice negative for ChR2 (total = 17 mice). Electrophysiology data was acquired on a Neuralynx (Neuralynx, Bozeman MT) system with a 32 kHz sampling rate. Full sampling was used to record spikes, detected with a manually set voltage threshold. LFPs were collected with a highpass filter between 0.1 and 0.3 Hz and a lowpass between 2000 and 9000 Hz. EMG was collected with a highpass filter of 10 Hz to prevent data saturation. All electrophysiology data was exported to MATLAB (Mathworks, Natick MA), and LFPs and EMGs were then lowpass filtered offline at 500 Hz and downsampled to 1000 Hz sampling rate. Spike sorting was performed with custom software (Simpleclust, http://github.com/open-ephys/simpleclust), using standard waveform features to classify spikes. Spikes that could not be assigned to a well-defined cluster were labeled as multi-unit activity, and triphasic waveforms were excluded as fibers of passage. Awake recordings were carried out in either a head-fixed setup or in a clear plastic bowl. Anesthetized recordings were performed with isoflurane in 100% oxygen, in which drug levels were increased if the animal showed any signs of motion, and decreased when the EEG showed burst suppression, for an average range of 0.6% to 1% isoflurane. Experiments in anesthetized trials were performed only after anesthesia was induced with at least 1.5% isoflurane and mice had lost the righting reflex, and isoflurane was maintained at at least 0.5% isoflurane throughout the stimulation period. Anesthetic levels were varied manually to stay within a lightly anesthetized range, by decreasing levels if the EEG showed burst suppression and increasing levels if mice showed any sign of movement. In sessions with automated motion quantification, two video cameras were mounted at two orthogonal angles to enable automated motion capture.

## Laser stimulation

ChR2 expressing neurons were activated with a DPSS laser with a wavelength of 473 nm. Halorhodopsin expressing neurons were activated with a DPSS laser with a wavelength of 579 nm. In VGAT-ChR2 mice, light was delivered as 30 s stimulation periods using steady light levels (DC stimulation), followed by at least 30 s (typically 60-–90 s) with no stimulation. Light was maintained at constant levels throughout a single 30 s period. For experiments comparing different laser strengths, the laser output was varied within a single session, but not within a single 30 s stimulation period. Simulations for the transmission of light through tissue at these different laser strengths were performed using the calculator developed by the Deisseroth lab (http://web.stanford.edu/group/dlab/cgi-bin/graph/chart.php). In VGAT-Cre mice, two sites were stimulated simultaneously by using a splitter (Doric

Lenses) to generate two matched light sources. Light levels were kept below 4 mW for all recordings except those mice with viral injections (*Figure 1—figure supplement 3–4*, *Figure 4d*), in which power was increased to between 4–5 mW to compensate for the lower expression levels. Most recording sessions in VGAT-Cre mice with viral injections used a 5 s DC stimulation period instead of 30 s as a precaution to avoid any tissue heating from the increased laser power, but the subset of sessions with 30 s stimulation periods showed similar results. Recording sessions were limited to no more than 60 stimulation trials to prevent habituation effects, although none were observed in the data, with typical sessions lasting approximately 2 hr.

## Histology

Animals were perfused using 4% paraformaldehyde (PFA) in phosphate-buffered saline (PBS) and brains were extracted and fixed in PFA-PBST. A vibratome was used to collect 60 micron coronal slices stored in PBS. Images show DAPI staining in blue and EYFP in green. 2x and 10x images were taken on a Zeiss Axio M2 microscope. In addition to the histology from mice in which we performed electrophysiological recordings (*Figure 1—figure supplement 1,2,5*; *Figure 4—figure supplement 2*), we also injected an additional VGAT-Cre mouse with the same halorhodopsin virus and performed histology at higher resolution. The same histological methods were used, except that slices were 65 microns thick. These images were taken on a Zeiss 750 confocal microscope (*Figure 4—figure supplement 3*). Cell counting was performed manually, and the proportion of cells that were positive (i.e., were surrounded by a fluorescent ring) was reported with error bars indicating the 95% confidence intervals calculated theoretically from the binomial distribution.

## Spectral analysis

Spectra were computed with the Chronux toolbox using 19 tapers over 30 s windows. Spectrograms were computed with 5 tapers in 5 s sliding windows every 1 s. Normalized spectrograms were computed by first taking the median power across all trials, and then dividing the power in each frequency by the mean of the power at that frequency during the 30 s pre-stimulus window. Error bars across multiple sessions (e.g. , *Figure 1d*) show the standard error of the mean across all trials. Statistical testing was done by taking the sum of power within a band of interest on individual trials, and then comparing power during the 30 s stimulation against power during the 30 s immediately preceding TRN stimulation. Change in power is reported as the median effect size and 95% confidence intervals, computed by inverting the Wilcoxon signed-rank test. Automatic slow wave event detection was performed by bandpassing the LFP between 0.1–5 Hz using a finite impulse response filter, detecting local minima with magnitude >100 µV, and computing the difference between the negative trough and subsequent peak. The top 3% percentile of peak-to-peak amplitudes within each session were selected as slow wave events. Any peak within a 400 ms window with standard deviation > 2000 was rejected as artifact. The difference in the number of peaks was statistically tested using a Wilcoxon signed rank test to compute the median difference in slow wave event numbers in the stimulated versus baseline periods for each trial. For analyses across electrodes, the change in power was computed for each electrode and the Bonferroni correction was applied for multiple comparisons across electrodes. To compare the number of activated electrodes in local versus global electrodes, and across laser power conditions, we treated the number of significant electrodes as a binomial distribution. We assumed a uniform prior for the binomial parameter, obtaining a beta density as the posterior distribution for each proportion. We estimated the difference between two conditions by sampling 1000 times from the posterior distributions in each condition, and calculating the median and 95% confidence intervals as the 2.5th and 97.5th percentiles of the difference between each resampled datapoint (i.e., a Monte Carlo bootstrap for the difference between two groups). To compute the PLV, each channel was filtered between 1-–4 Hz and Hilbert transformed to extract the instantaneous phase. The PLV was then taken as *the* circular mean of the difference in phase for each electrode relative to the electrode closest to the optical fiber (*Lachaux et al., 1999*). To test whether the magnitude of the PLV was greater than expected by chance, the trial labels were shuffled 500 times and the true PLV was compared to permuted PLV values from the shuffled trials, at alpha = 0.05. To report the confidence interval for the angle of the PLV, the trials were resampled with a bootstrap procedure and the PLV angle across the resampled data was calculated 500 times.

## Behavioral analysis

All trials used for behavioral analysis were collected while mice behaved freely in a clear plastic bowl. Recording sessions lasted 1–2 hr and were performed during the day. Two video cameras were mounted at two orthogonal angles to enable automated motion capture. EMG effects were calculated using the Chronux toolbox to determine power in the 10–200 Hz band in non-overlapping windows of 1 s width. Power was summed across all frequencies within the band to obtain a single measure of EMG power. Statistical testing was performed with the Wilcoxon signed-rank test, comparing EMG power within each laser trial with the EMG power in the associated pre-stimulus period. The rapid onset of the EMG effect was assessed by comparing EEG power in the second prior to laser onset with power in the second following laser onset. Overall changes in behavior were calculated by comparing the [-30 0] baseline period to the [0 30] laser-induced period.

The correlation between EEG and EMG power was calculated by computing the correlation coefficient between the change in delta (1-–4 Hz) power in the EEG, calculated as the difference in the [-30 0] and [0 30] periods relative to laser onset, and correlating it with the change in EMG power across those same periods. Statistical significance was tested by performing a bootstrap with 1000 iterations on the paired power data, taking the 97.5% percentile of these resampled values, and testing whether its absolute value was larger than the correlation computed on a randomly shuffled set of times. This provided a test at significance level $\alpha = 0.05$ of whether the correlation between EEG and EMG effects was significantly higher during laser trials than would be expected during baseline conditions. Automated motion scoring was computed using automated custom software written in Matlab (http://github.com/jvoigts/optical_flow_analysis) that calculated the optical flow for each frame in the video via the Horn–Schunck method (*Horn and Schunck, 1981*). The points of maximal motion in each camera view were used to compute a motion vector in the horizontal plane. The motion vector was normalized by its mean to avoid artifact due to variations in lighting conditions and camera placement. The magnitude of the motion vector for each frame was then smoothed using a moving average filter (sigma = 200 frames/6.67 s) and used as a proxy for the magnitude of animal's overall motion. In order to ensure objective assessment of sleep, semi-automated sleep scoring was performed using an algorithm that first detects wake states as periods with heightened muscle tone by computing the instantaneous amplitude between 60–200 Hz of the EMG or cranial screw recording motor activity, smoothing with a Gaussian filter of 50 ms, and then using a manually entered threshold to identify wake states as those with at least 5 s of data above the threshold. Second, it computes the ratio of <4 Hz and 4–16 Hz power in the EEG, smooths with a Gaussian filter, and uses a manually entered threshold to segment non-REM and REM states. Sessions where the automated algorithm could not achieve a separation of sleep and wake states were not included in the analysis (1 session out of 13 total was excluded). The user selecting the EEG and EMG thresholds was blind to the timing of laser stimulation during the sleep scoring procedure. As with spectral effects, changes were reported as the median power change in dB, and 95% confidence intervals were computed by inverting the Wilcoxon signed-rank test. The shuffled control to test whether halorhodopsin-driven decreases in slow wave activity were larger than would be expected by chance was performed by pseudorandomly selecting an equivalent number of laser onset times during spontaneous NREM sleep and recalculating the change in slow wave power. This shuffling procedure was repeated 400 times to produce confidence intervals.

## Single unit analysis

Statistical comparisons of firing rates were performed using confidence intervals derived from the Wilcoxon signed-rank test to quantify the median difference between each neuron's pre-stimulus ([-30 0] s) and stimulus-induced ([0 30] s) spike rates. To compute phase modulation, the instantaneous phase was calculated from the local LFP channel by bandpass filtering the LFP between 1–4 Hz with a finite impulse response filter, taking the Hilbert transform, and then extracting the angle. The phase distribution of individual units relative to the delta phase was quantified using the modulation index (MI), adapted from phase-amplitude measurements (*Tort et al., 2010*), which measures the Kullback-Liebler distance between the observed phase distribution and a uniform distribution. The MI for spikes was computed over 10 phase bins as $\sum_{i=1}^{10} p_i \log_2 p_i + \log_2 10$, where $p_i$ is the proportion of spikes falling within a given phase bin. The MI for gamma power was computed over 100 phase bins using a

sinusoid fit to model the amplitude of the gamma oscillation. The effect of TRN activation on unit phase modulation was assessed as in the firing rate case, with median effect sizes and 95% confidence intervals derived from a Wilcoxon signed-rank test comparing each neuron's MI values in the [-30 0] and [0 30] periods. When comparing cortical units on channels with and without a slow wave effect, normalized delta power changes were computed by dividing delta (1–4 Hz) power by total 0–50 Hz power in the [-30 0] and [0 30] periods. Electrodes with a normalized delta power increase of at least 2% during TRN activation were labeled as having a delta effect. Subcortical units were divided into two categories based on the time between the peak and trough of the waveform. Narrow (<200 ms peak-to-trough) units were further subdivided into three categories based on their spike rate response to the laser, computed by taking the difference of their spike rates in the [0 30] s period vs. the [-30 0] period. Phase modulation was computed relative to the local LFP for each unit. To account for sign reversals due to electrode placement or referencing and ensure consistent phase measurements across units, thalamic LFPs were flipped such that the laser-induced deflection was negative across all channels. The fraction change in phase-locking strength for each unit was calculated by subtracting the MI during the stimulated period from the MI during baseline, and then normalizing by the MI during baseline. The peak phase was selected by dividing spikes into 10 phase bins and identifying the bin containing the most spikes. The narrowness of the phase distribution was tested by computing the kurtosis separately for putative TC and putative RE units using the CircStat toolbox (*Berens, 2009*). The difference between the two unit types was then calculated. To test whether this difference was significant, the difference in kurtosis was bootstrapped 1000 times with random resampling of units, shuffling the unit type assignment, and then the original difference in kurtosis was compared to the 95% confidence interval derived from the 2.5th and 97.5th percentile of the resampled differences. To test the timing of the rate decrease relative to laser onset, 10 ms windows in the 2 s pre-stimulus period were used to create a distribution of baseline values, which was bootstrapped 1000 times to determine a threshold for significant change (the 0.25th percentile, $\alpha = 0.005$). 10 ms windows after laser onset were then compared to this threshold to assess timing of a significant change. Triggered LFP analysis was done analogously, averaging the mean LFP value in 10 ms bins. For LFP and spike timing analyses, the alpha level was set at 0.005 to correct for multiple comparisons across time bins (10 bins of 10 ms width to span the 100 ms deflection interval).

## OFF period analysis

To detect OFF periods, we combined all multi-unit and single-unit activity on a single channel into a point process representation, and then smoothed with a Gaussian kernel with a standard deviation of 20 ms to approximate an instantaneous firing rate in the units surrounding that electrode. OFF periods were labeled as any period of at least 50 ms with a firing rate of zero. To verify that OFF periods were occurring at a greater rate than would happen by random chance, we also computed OFF periods on simulated data with the same mean firing rate as the experimental data. The simulated data was generated by taking the interspike intervals throughout the recording period, fitting a gamma distribution to these intervals, and then generating a new spike train from that gamma distribution with the same number of spikes as the original dataset. The OFF periods were then calculated with the same method for the simulated data. Statistical testing for OFF periods was performed across trials within each session: the percent of time spent in an OFF period during laser stimulation was compared to the percent of time spent in an OFF period in the 30 s preceding laser stimulation with the Wilcoxon signed-rank test. The significance of this difference within each animal is reported. The percent of time in OFF periods was compared to simulated data by running the simulation 1000 times and testing whether the experimental value was greater than the 97.5th percentile of the simulated value. Statistical testing for the phase distribution of OFF periods was computed by splitting the data into 10 phase bins and testing for uniform distribution of OFF periods using Pearson's chi-square test – a significant result indicated that OFF periods were not uniformly distributed across the LFP slow wave phases, but rather appeared predominantly at specific phases.

## GLM analysis

Temporal firing rate patterns were quantified using a generalized linear model, in which a unit's spike rate over time was modeled as a point process with rate as a function of previous spike history (*Truccolo et al., 2005*). The model covariates consisted of either a 1 or a 0 to indicate whether a

spike was observed in any given preceding time bin. The model used 50 bins of 2 millisecond width each. GLMs were fit in Matlab and confidence intervals were calculated using 'glmfit', which were then used to determine for each cell whether the parameter estimate for the [2 4] ms was significantly different during TRN stimulation as compared to baseline, with $\alpha = 0.05$. The proportion of cells with a significant change in model parameter estimates is reported.

## Acknowledgments

This work was funded by NIH TR01 GM104948 to ENB, K99 NS078115 to MMH, and Canadian Institutes of Health Research and Harvard Society of Fellows fellowships to LDL. We thank Christopher Moore for helpful discussions. We are grateful to Ralf Wimmer, Rodrigo Garcia, and Jenny Pei for advice and assistance with experimental techniques.

## Additional information

### Competing interests

ENB: Reviewing editor, *eLife*. The other authors declare that no competing interests exist.

### Funding

| Funder | Grant reference number | Author |
| --- | --- | --- |
| National Institutes of Health (NIH) | TR01 GM104948 | Emery N Brown |
| Canadian Institutes of Health Research (CIHR) | Doctoral Research Award | Laura D Lewis |
| Society of Fellows, Harvard University | Junior Fellowship | Laura D Lewis |
| National Institutes of Health (NIH) | K99 NS078115 | Michael M Halassa |

The funders had no role in study design, data collection and interpretation, or the decision to submit the work for publication.

### Author contributions

LDL, JV, Research conception and design, acquired data, analyzed data, wrote article, reviewed and revised article.; FJF, Acquired data, reviewed and revised article; LIS, Data acquisition, data analysis, revising article; MAW, Research design, reviewed and revised article; MMH, Research conception and design, acquired data, reviewed and revised article; ENB, Research conception and design, reviewed and revised article

### Ethics

Animal experimentation: All experimental procedures were approved by the MIT Committee on Animal Care (protocol number #0514-038-17).

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
