## [Decision Letter]

Thank you for submitting your work entitled "Thalamic reticular nucleus induces fast and local modulation of arousal state" for peer review at *eLife*. Your submission has been favorably evaluated by Eve Marder (Senior Editor), Michael Frank (Reviewing Editor), and two reviewers.

The reviewers have discussed the reviews with one another and the Reviewing editor has drafted this decision to help you prepare a revised submission.

This study investigates the circuit mechanisms to generate spatially isolated slow waves in the cortex. Using optogenetics, electrophysiology, EEG and EMG recordings, the authors showed that local tonic activation of the thalamic reticular nucleus (TRN) rapidly induced slow wave (1–4 Hz) in isolated cortical regions (or more globally depending on stimulation intensity). They also provide some limited evidence that inhibiting TRN during sleep reduced slow wave activity. Single unit recording indicated that tonic TRN simulation predominantly suppressed thalamic firing rates, while at the same time promoting thalamic bursting. Importantly, the induced slow wave was behaviorally relevant, associated with a progressive reduction in arousal state, reduced movement. The main conclusion is that TRN-mediated thalamic inhibition can serve as a mechanism for local modulation of cortical arousal state. This study addresses a critical topic in thalamic control of cortical state, and is well motivated by a series of physiology and behavioral studies that suggest TRN is the key mediator of thalamocortical dynamics implicated in sensory processing, attentional gating and arousal/sleep state transition.

Essential revisions:

Overall, this is a well-designed study and the experiments are carefully performed, which could potentially contribute multiple important findings, as outlined above. However, more information and experimental data are needed in certain aspects of the study to fully address the question being investigated. There are also some issues with presentation of the data.

1) The channelrhodopsin data were substantially more fleshed out than those for halorhodopsin, which we feel should be held to the same standard. The halo study lacks control recordings from TRN and relay neurons to show what this manipulation does to thalamic activity, and the data provided are very limited in scope (measures of cortical delta power only) and marginally significant (Figure 4, right) with small effects. Yet this inactivation study is a key experiment to support the conclusions in the subsection “Circuit mechanisms underlying the induced thalamocortical slow waves” that "slow waves in the intact brain require the involvement of cortical, TRN, and TC neurons in a coordinated rhythm" and in the Abstract that "inhibiting TRN during sleep reduces slow waves". The existing experiment on cortical LFP's is somewhat helpful but too sparsely presented to be convincing. It's essential to show that the virus expressed well, that it expressed in the targeted cells and not in unintended cells, and importantly, that halorhodopsin was effective, i.e. that it actually did suppress TRN spiking. This requires the combination of strong viral transduction and effective optical stimulation in the right place. For interpreting the experiment in the context of the hypotheses, it would also be very helpful to record from TC relay cells under halo conditions, since those are the neurons that can actually communicate with the cortex where LFPs are being recorded. All but the TC recordings are basic control experiments. While nearly all of these steps were conducted for the main experiment using ChR2 stimulation of TRN cells, the NpHR (halo) experiments require their own controls.

2) The study showed that higher strength of laser power induced slow waves in a more widespread region of the cortex, but the underlying mechanism was not examined. Was it because higher laser power inhibited more thalamic nucleus projecting to a wider cortical area, or because stronger slow wave traveled more widespread across the cortex? Analyzing the slow wave data in different electrodes may solve this puzzle.

3) Please provide more information on slow wave characteristics and how they were affected by TRN activation. What was the amplitude criterion for slow waves? How many slow waves were detected in normal wakefulness (without TRN activation)? Were more, stronger and/or steeper slow waves detected after TRN activation compared to non-stimulated wakefulness? These characteristics can't be simply obtained from the delta power.

4) In subcortical recording, 6–8 microdrives (1–3 stereotrodes each) were implanted in each of five mice, but only 28 units were isolated in total. The number was relative small given the number of stereotrodes used. Please explain. Also, the distribution of spike waveforms is very clean with a clear cut between the 'narrow' and 'wide' spike waveforms in Figure 3. This raises the question of whether those 28 units were selectively included or not. Please provide sample waveforms for putative TRN and TC neurons.

5) Though the effect of TRN stimulation on slow waves during anesthesia was reported, the impact of tonic TRN stimulation during natural sleep was not examined. Though both anesthesia and natural sleep represent a state of decreased arousal, the underlying thalamocortical dynamics could be different. One important question is whether tonic TRN stimulation during NREM sleep would increase or decrease spindle oscillations. To get a full picture of TRN activation in slow wave modulation, the effect of TRN stimulation during natural sleep needs to be reported, or the conclusions tempered/ the issue discussed.

6) It has been shown that during the behavioral transition from wakefulness to sleep, the firing patterns of reticular neurons undergo a shift from tonic firing (20–40 Hz) during arousal, to lower frequencies (10–20 Hz) during drowsiness, and to rhythmic spike bursts during synchronized slow wave sleep (56). Somewhat counterintuitively, this study shows that tonic activation of TRN induces sleep-like behavior. The implication of such discrepancy needs to be discussed.

7) One major conclusion of this study is that TRN could serve as a circuit mechanism to modulate local slow waves in the cortex. However, in this study, only the somatosensory sector of TRN was simulated. It is not clear why the somatosensory sector of TRN was targeted and whether other TRN regions have been simulated. Considering the topographically mapped projections from thalamus to cortex, it is important and desirable to know whether simulating a different TRN region would induce slow waves in other cortical regions. This issue is particular relevant in light of recent finding from the authors that the TRN composes of distinct subnetworks engaged in different functional circuits (25). Thus while we do not expect yet other experiments to stimulate other thalamic nuclei, it would be helpful to discuss this issue and to qualify the sweeping statements about TRN specifically throughout.

[Editors' note: further revisions were requested prior to acceptance, as described below.]

Thank you for resubmitting your work entitled "Thalamic reticular nucleus induces fast and local modulation of arousal state" for further consideration at *eLife*. Your revised article has been favorably evaluated by Eve Marder (Senior Editor), Michael Frank (Reviewing Editor), and two reviewers. The manuscript has been improved but there are some remaining issues that need to be addressed before acceptance, as articulated clearly and concisely by the reviewers so there is no need to summarize them further.

*Reviewer #1:* I remain excited about this paper but also note that I remain a bit surprised by the findings and expect this to be a controversial paper. This would not necessarily be a bad thing since the slow oscillation community is overall so much focused on neocortex that this current paper may help to generate some new research directions and models.

Overall, the authors have thoroughly addressed most of the points I had raised. I note the fact that they chose not to perform the electrophysiology experiment we had requested. I would like to leave this up for discussion / decision by the editor. I have few last requests that are easy to address:

Concerning the traveling waves analysis, was there a correlation between electrode distance to optical fiber and the calculated phase offset? Can you please add a scatterplot to Figure 1—figure supplement 8 with electrode distance as X-axis and phase lag as Y-axis?

*Reviewer #2:* The revised paper has significantly improved. I have just two additional comments:

1) The Abstract says: "we show that local tonic activation of thalamic reticular nucleus (TRN) rapidly produces slow wave activity in a spatially restricted region of cortex". It's a small point, but I think "produces" misstates the phenomenon. The TRN may "induce", "trigger", or "facilitate" slow waves, but it does not produce them; the cortex does (as the authors' first Abstract sentence implies).

2) Demonstration of NpHR expression and efficacy in TRN: The authors argue that it is enough to show histologically (Figure 4—figure supplement 2) that their viral transduction worked and was effective on slow waves. My opinion is that this is below the usual standards in the field, but I appreciate that actual thalamic recordings during halorhodopsin activation (as they managed to do for ChR2) would be difficult. The meager histological evidence provided is not very compelling, however. As the authors point out themselves, there is good evidence for neuronal heterogeneity in TRN. The single, small, supplementary histological photo does not allow one to judge whether NpHR expression was generalized to all TRN cells in an infected sector, or whether there was cellular selectivity. Such evidence would never suffice, for example, to demonstrate efficacy of NpHR expression in cortical interneurons where subtypes are diverse and well characterized. Cell counts and double-labeling are the norm. Different viral serotypes can have very strong cellular tropisms and lead to selective expression. To my eye, in fact, the expression in the supplementary image appears very patchy and nonuniform. TRN neurons are well known to interact via inhibitory synapses but the specific intra-TRN circuits are unknown, so patchy expression may not be a good predictor of the effect of optogenetic suppression on overall TRN activity.

So, a modest request: can the authors strengthen their case a bit by looking at NpHR expression patterns more closely, at higher power and resolution, and report whether transduction seems to be generalized or selective across TRN neurons?

---

## [Author Response]

*Essential revisions: Overall, this is a well-designed study and the experiments are carefully performed, which could potentially contribute multiple important findings, as outlined above. However, more information and experimental data are needed in certain aspects of the study to fully address the question being investigated. There are also some issues with presentation of the data. 1) The channelrhodopsin data were substantially more fleshed out than those for halorhodopsin, which we feel should be held to the same standard. The halo study lacks control recordings from TRN and relay neurons to show what this manipulation does to thalamic activity, and the data provided are very limited in scope (measures of cortical delta power only) and marginally significant (Figure 4, right) with small effects. Yet this inactivation study is a key experiment to support the conclusions in the subsection “Circuit mechanisms underlying the induced thalamocortical slow waves” that "slow waves in the intact brain require the involvement of cortical, TRN, and TC neurons in a coordinated rhythm" and in the Abstract that "inhibiting TRN during sleep reduces slow waves". The existing experiment on cortical LFP's is somewhat helpful but too sparsely presented to be convincing. It's essential to show that the virus expressed well, that it expressed in the targeted cells and not in unintended cells, and importantly, that halorhodopsin was effective, i.e. that it actually did suppress TRN spiking. This requires the combination of strong viral transduction and effective optical stimulation in the right place. For interpreting the experiment in the context of the hypotheses, it would also be very helpful to record from TC relay cells under halo conditions, since those are the neurons that can actually communicate with the cortex where LFPs are being recorded. All but the TC recordings are basic control experiments. While nearly all of these steps were conducted for the main experiment using ChR2 stimulation of TRN cells, the NpHR (halo) experiments require their own controls.*

We have now added histology as a supplement to Figure 4. We demonstrate that the NpHR virus expressed selectively in TRN (Figure 4—figure supplement 2), analogously to the ChR2 virus (Figure 1—figure supplement 5). Given the successful expression in TRN, the fact that we used a widely used virus that is known to express halorhodopsin, and the electrophysiological evidence that we suppressed slow wave activity, we believe that adding histological figures is sufficient to show that our optogenetic manipulation was selective and effective in TRN. Halorhodopsin has been previously well characterized (e.g. Zhang et al., Nature, 2007; Gradinaru et al., Brain Cell Bio, 2008), the virus has been used in several prior suppression experiments (e.g. Land et al., Nat. Neuro. 2014; Delevich et al., J. Neurosci. 2015; McCutcheon et al., Front. Neur. Circ. 2014) and we verified successful infection in TRN (the new Figure 4—figure supplement 2). Therefore, we have strong evidence for the selectivity and effectiveness of our manipulation by adding histology alone. Given the histological evidence of halorhodopsin expressing in TRN, and previous studies of halorhodopsin, there is no alternative hypothesis we can think of that could explain our results, other than successful suppression of TRN. Single unit recording in the TRN is challenging and is a lengthy process, requiring at least 6 months to record from even a small number of animals, so we were not able to acquire new unit recordings within the resubmission deadline. However, since we have added histology showing expression of halorhodopsin in TRN, and since many other studies have demonstrated the effects of the AAV halorhodopsin virus, the evidence for our manipulation is strong. We agree it would be interesting for future studies to examine the dynamics of unit firing during inhibition, as these changes may be complex rather than a simple decrease in firing rates (similar to the observed heterogeneity in our ChR2 manipulation), and future studies could combine slice experiments with computational models to explore those dynamics and provide insight into the microcircuitry of TRN. We believe these revisions fully address the reviewers’ request to show effective manipulations in TRN, as the new histological figure demonstrates successful and selective viral expression.

We have removed the statement about TRN inhibition from the Abstract to reduce its emphasis, as we agree with the reviewers that this paper primarily focuses on TRN stimulation. Our statement about the requirement of all three structures to produce a slow wave was not intended to refer to the halorhodopsin experiment. Rather, we intended to discuss the fact that studies lesioning thalamus have in general not produced slow waves; it instead seems that all three structures need to be intact for this circuit to oscillate. We have added a clarification to this statement to avoid linking it to the halorhodopsin findings, and instead discuss the point that physiological inhibition of thalamus, rather than complete suppression or lesioning, seems to play a key role in slow wave generation. The primary conclusion of our study is that tonic activation of TRN can decrease cortical arousal state, and that this modulation is rapid, local, and behaviourally relevant.

*2) The study showed that higher strength of laser power induced slow waves in a more widespread region of the cortex, but the underlying mechanism was not examined. Was it because higher laser power inhibited more thalamic nucleus projecting to a wider cortical area, or because stronger slow wave traveled more widespread across the cortex? Analyzing the slow wave data in different electrodes may solve this puzzle.* We appreciate the suggestion and have added new analyses showing the phase relationships between slow waves at distant cortical sites (subsection “TRN activation selectively controls a local ipsilateral cortical region”, second paragraph). While these data are not sufficient to decisively conclude the underlying mechanism, they suggest that dynamics across the recorded regions are significantly synchronized, with small but detectable phase delays across cortical sites. Inducing slow waves through TRN stimulation has very little effect on the magnitude or phase of cross-cortical dynamics (in the aforementioned paragraph, Figure 1—figure supplement 8). The lack of substantial changes in phase offsets or increased phase-locking argues against the hypothesis that stronger slow waves are induced locally but are traveling further across cortex, and would be consistent with the idea that thalamocortical loops are oscillating according to their intrinsic dynamics. However, our experiments cannot definitely verify this. Future studies could investigate this by recording in additional thalamic sites, or by disrupting slow waves in a local cortical region and examining the impact on distant regions. We discuss these possibilities in the text, and mention that this also demonstrates that TRN activation can generate slow waves with phase offsets, which have been observed in natural sleep and in anesthesia.

*3) Please provide more information on slow wave characteristics and how they were affected by TRN activation. What was the amplitude criterion for slow waves? How many slow waves were detected in normal wakefulness (without TRN activation)? Were more, stronger and/or steeper slow waves detected after TRN activation compared to non-stimulated wakefulness? These characteristics can't be simply obtained from the delta power.*

We primarily sought to report delta power because individually detecting slow waves yields variable results depending on the detection criteria. For example, if slow waves become smaller in amplitude, and no longer reach detection threshold, it can appear that slow waves occur less frequently when in fact they were as frequent but smaller. We also report the statistics of OFF periods, which are the neuronal basis for the LFP slow waves (Figure 2). However, we recognize that slow wave events are of interest, so we have added new analyses demonstrating slow wave event statistical properties during TRN stimulation. We now compare the frequency and amplitude of slow waves in stimulation vs. baseline, and show a substantial increase in the number of events during TRN stimulation (subsection “Tonic activation of TRN produces cortical slow waves”). We also report that the amplitude of the slow waves shifts to an asymmetric waveform in which the positive-going peak increases in amplitude, similar to the slow waves seen in natural sleep. We'd just like to caution that this type of analysis depends on the precise definition of a slow wave that is used, so most analyses we present still use delta power to provide a less biased metric of ongoing cortical dynamics. It is difficult to determine from LFP alone the true frequency and amplitude of slow waves, as there are no objective selection criteria. Additional relevant statistics are provided in our analysis of OFF periods (which analyzes periods with no unit firing rather than LFP slow waves) comparing the duration of OFF periods during stimulation vs. wake (subsection “Cortical units rapidly phase-lock to induced slow waves and undergo OFF periods”), which quantifies the neuronal basis of the induced LFP slow waves.

*4) In subcortical recording, 6–8 microdrives (1–3 stereotrodes each) were implanted in each of five mice, but only 28 units were isolated in total. The number was relative small given the number of stereotrodes used. Please explain. Also, the distribution of spike waveforms is very clean with a clear cut between the 'narrow' and 'wide' spike waveforms in Figure 3. This raises the question of whether those 28 units were selectively included or not. Please provide sample waveforms for putative TRN and TC neurons.*

The anatomy of TRN makes it challenging to record from: it is an extremely thin structure located deep in the brain, and electrodes reach it only after gradual lowering of the electrodes on many successive days, with the entire lowering process taking at least a week. Some electrodes get stuck during this process, and remain trapped in cortex or superficial structures, and some electrodes become bent or their quality degrades. The number of units we report here is very typical given the challenges of recording from small subcortical structures in non-anesthetized animals (e.g. Halassa et al., Nat Neuro, 2011: 11 TRN units in 3 mice; Bartho et. al, Neuron, 2014: 17 TRN units in 5 rats).

The only selection performed on the isolated units was to exclude units whose waveforms showed three peaks (as described in the Methods section), as these may correspond to fibers of passage rather than local neuronal firing. No other selection was performed on the units. We should also have clarified in the main text that only four of these mice had complete microdrive implants; the fifth mouse’s thalamic units were recorded by attaching stereotrodes directly to the optical fiber, positioning them precisely at the site of stimulation but resulting in a smaller number of units (and only TC units), and we have add a more detailed section on this in the Methods (subsection “Cortical implants”). We agree that the histogram has an unusually clean separation between the two groups; this is likely a chance occurrence and also reflects the fact that not very many units were recorded, so it is fairly statistically likely to observe a gap in a binned histogram because we don’t have extensive sampling of units. We have added the requested sample waveforms (Figure 3—figure supplement 1) to illustrate the range of waveform shapes we observed.*5) Though the effect of TRN stimulation on slow waves during anesthesia was reported, the impact of tonic TRN stimulation during natural sleep was not examined. Though both anesthesia and natural sleep represent a state of decreased arousal, the underlying thalamocortical dynamics could be different. One important question is whether tonic TRN stimulation during NREM sleep would increase or decrease spindle oscillations. To get a full picture of TRN activation in slow wave modulation, the effect of TRN stimulation during natural sleep needs to be reported, or the conclusions tempered/ the issue discussed.*

We have added a new section focusing on the effect of TRN during natural sleep (”Tonic TRN stimulation further increases slow wave activity during sleep”). TRN stimulation increased slow wave power during natural NREM sleep (Figure 4). We have also added new statistics to demonstrate that it does not increase spindle power – in fact, it induces a slight decrease in power likely due to the increased slow wave amplitude and induction of prolonged OFF periods. It’s interesting that tonic TRN stimulation induces only slow waves and not spindles, even though phasic TRN stimulation during sleep can cause spindles (26), and we discuss this further in the Discussion (subsection “TRN supports both slow waves and sleep spindles”).

*6) It has been shown that during the behavioral transition from wakefulness to sleep, the firing patterns of reticular neurons undergo a shift from tonic firing (20–40 Hz) during arousal, to lower frequencies (10–20 Hz) during drowsiness, and to rhythmic spike bursts during synchronized slow wave sleep (56). Somewhat counterintuitively, this study shows that tonic activation of TRN induces sleep-like behavior. The implication of such discrepancy needs to be discussed.*

We have added a new Discussion section focusing on this point and previous studies reporting unit recordings in TRN (“Neuronal activity in TRN across arousal states”). In part due to the challenges of recording units in TRN, only a small number of studies have reported TRN firing patterns during sleep. There is increasing evidence that more than one cell type and firing rate pattern exists in TRN, and the local circuitry may cause our stimulation to only increase firing in some subset of TRN neurons. For instance, in Figure 4 of [56], they report that a subset of the TRN neurons they recorded had higher firing rates during NREM sleep than during wake, despite their switch into a rhythmic bursting mode. This is consistent with our data, in which only some neurons increase their firing rates, and both stimulated and suppressed neurons become phase-locked to the slow waves (e.g. Figure 3). Although our stimulation paradigm is tonic, the resulting firing pattern is locked to slow waves, indicating that tonic drive to TRN is sufficient to induce oscillatory patterns within TRN as well. We therefore suggest that our results do not represent a discrepancy with past electrophysiology studies, but rather is consistent with the finding that during natural sleep one can observe high TRN firing rates that are intermittent, as the TRN units are firing at high rates at only a limited phase of the slow wave cycle, and that the firing patterns within TRN are heterogeneous and different subsets of neurons may play different functional roles in slow wave generation.

*7) One major conclusion of this study is that TRN could serve as a circuit mechanism to modulate local slow waves in the cortex. However, in this study, only the somatosensory sector of TRN was simulated. It is not clear why the somatosensory sector of TRN was targeted and whether other TRN regions have been simulated. Considering the topographically mapped projections from thalamus to cortex, it is important and desirable to know whether simulating a different TRN region would induce slow waves in other cortical regions. This issue is particular relevant in light of recent finding from the authors that the TRN composes of distinct subnetworks engaged in different functional circuits (25). Thus while we do not expect yet other experiments to stimulate other thalamic nuclei, it would be helpful to discuss this issue and to qualify the sweeping statements about TRN specifically throughout.*

We completely agree with this point and have edited the manuscript to include the caveat that we focused our stimulation on specific regions of TRN, and to be more clear about the regions that were stimulated. It is likely that during our high-power (global) stimulation, we stimulated more than just the somatosensory sector of TRN, as light is expected to spread in this case and reach other sectors. We have added a supplemental figure estimating the spread of the light from the somatosensory sector to illustrate that we likely stimulated other sectors as well when stimulating at high laser power (Figure 1—figure supplement 7). We have also added some discussion of this issue (subsection “Neuronal activity in TRN across arousal states”), and it would be interesting in future studies to further explore the mapping of TRN sectors.

[Editors' note: further revisions were requested prior to acceptance, as described below.]

Reviewer #1:

*I remain excited about this paper but also note that I remain a bit surprised by the findings and expect this to be a controversial paper. This would not necessarily be a bad thing since the slow oscillation community is overall so much focused on neocortex that this current paper may help to generate some new research directions and models. Overall, the authors have thoroughly addressed most of the points I had raised. I note the fact that they chose not to perform the electrophysiology experiment we had requested. I would like to leave this up for discussion / decision by the editor. I have few last requests that are easy to address:*

*Concerning the traveling waves analysis, was there a correlation between electrode distance to optical fiber and the calculated phase offset? Can you please add a scatterplot to Figure 1—figure supplement 8 with electrode distance as X-axis and phase lag as Y-axis?*

We have added this new figure as a supplement to Figure 1 (Figure 1—figure supplement 9), showing that there is no significant correlation between distance and phase offset in our data. This result could indicate that waves do not travel, and is consistent with the idea of independent thalamocortical oscillators, as is suggested by Figure 1—figure supplement 8. It could also reflect the fact that many regions that are physically distant are nevertheless synaptically close (e.g. bilateral somatosensory cortex might be expected to have a smaller phase lag than ipsilateral frontal and somatosensory cortices, despite similar electrode distances), and traveling waves could conceivably spread in complex patterns due to these anatomical connectivity patterns. This figure does not definitively demonstrate absence of a correlation, and future work could potentially identify in greater detail the spatial structure of induced wave spread and the relative contributions of thalamocortical vs. corticocortical drive.

Reviewer #2:

The revised paper has significantly improved. I have just two additional comments:

1) The Abstract says: "we show that local tonic activation of thalamic reticular nucleus (TRN) rapidly produces slow wave activity in a spatially restricted region of cortex". It's a small point, but I think "produces" misstates the phenomenon. The TRN may "induce", "trigger", or "facilitate" slow waves, but it does not produce them; the cortex does (as the authors' first Abstract sentence implies).

We have replaced this statement with 'induces'.

*2) Demonstration of NpHR expression and efficacy in TRN: The authors argue that it is enough to show histologically (Figure 4—figure supplement 2) that their viral transduction worked and was effective on slow waves. My opinion is that this is below the usual standards in the field, but I appreciate that actual thalamic recordings during halorhodopsin activation (as they managed to do for ChR2) would be difficult. The meager histological evidence provided is not very compelling, however. As the authors point out themselves, there is good evidence for neuronal heterogeneity in TRN. The single, small, supplementary histological photo does not allow one to judge whether NpHR expression was generalized to all TRN cells in an infected sector, or whether there was cellular selectivity. Such evidence would never suffice, for example, to demonstrate efficacy of NpHR expression in cortical interneurons where subtypes are diverse and well characterized. Cell counts and double-labeling are the norm. Different viral serotypes can have very strong cellular tropisms and lead to selective expression. To my eye, in fact, the expression in the supplementary image appears very patchy and nonuniform. TRN neurons are well known to interact via inhibitory synapses but the specific intra-TRN circuits are unknown, so patchy expression may not be a good predictor of the effect of optogenetic suppression on overall TRN activity. So, a modest request: can the authors strengthen their case a bit by looking at NpHR expression patterns more closely, at higher power and resolution, and report whether transduction seems to be generalized or selective across TRN neurons?*

We have now added new high-resolution histology figures and cell counting to more closely examine the NpHR expression patterns, and report that transduction appears to be relatively generalized in TRN (Figure 4—figure supplement 3). The reason for the patchy appearance of the expression pattern is due to the anatomical structure of TRN. TRN is a weblike ('reticulated') structure that appears patchy because it is perforated with axon bundles passing through the structure (i.e. the TRN itself has a patchy structure). The new images show more clearly the distribution of EYFP and cell bodies, with higher resolution microscopy and images of individual cell bodies (Figure 4—figure supplement 3). The EYFP is membrane-bound, so it is absent within the cell bodies, and expression is seen as a ring of fluorescence that surrounds the cell body (Figure 4—figure supplement 3, left panels). While some fluorescence can be detected outside the TRN, this reflects expression in projections from TRN rather than in cell bodies outside the target region (Figure 4—figure supplement 3, bottom panels, no rings of fluorescence appear around thalamic cell bodies). The cell bodies surrounded by a ring of fluorescence appear only in TRN. We have also now added cell counting to show that the majority of TRN neurons were positive (Figure 4—figure supplement 3). We were not able to record new units within the timeframe required by *eLife*, as we were limited to 2 months and the process of microdrive construction, implantation, electrode lowering, and recording in multiple mice takes longer than that, but we completely agree it would be interesting for future studies to examine the dynamics within TRN during inhibition. We have added discussion of the histology in the text, and we state only that we achieved 'partial inhibition' of TRN, as we do not infect the entire structure and are also limited by the spread of the light. However, within the stimulated region, expression appears to be fairly widespread (Figure 4—figure supplement 3). We have also added this caveat mentioned by the reviewer to the Discussion (subsection “TRN as one element of arousal control”), highlighting that different expression patterns could indeed potentially alter the results.